# ZEROTH-ORDER SHARPNESS-AWARE LEARNING WITH EXPONENTIAL TILTING

## ABSTRACT

Classic zeroth-order optimization approaches typically optimize for a smoothed version of the original function, i.e., the expected objective under randomly perturbed model parameters. This can be interpreted as encouraging the loss values in the perturbation set to be small on average. Popular sharpness-aware minimization (SAM) objectives, however, typically focus on the largest loss within the neighborhood to arrive at flat minima more effectively. In this work, we connect zeroth-order optimization (and its corresponding objectives) with SAM approaches explicitly, grounded by an exponential tilting objective that provides a natural transition between the `average` and the `max`. We explore new zeroth-order algorithms to solve a *soft* SAM objective parameterized by a tilting parameter $t$ that covers the average and min-max formulation as special cases, as well as precise characterizations of the sharpness notions of the tilted SAM framework. Practically, our approach can be used as a gradient-free and memory-efficient alternative to SAM variants, and it achieves better generalization compared to vanilla zeroth-order baselines on a wide range of downstream tasks, including classification, multiple choice QA, and language generation.

## 1 INTRODUCTION

Zeroth-order optimization has gained traction when the first-order or higher-order gradient access is unavailable, unreliable, or expensive. Applications include black-box adversarial attacks (Chen et al., 2017), fine-tuning large language models (Chen et al., 2023; Malladi et al., 2023; Zhang et al., 2024), differentially private learning (Zhang et al., 2023; Tang et al., 2024), and science problems. Consider the standard empirical risk minimization (ERM) problem: $f(x) := \frac{1}{N} \sum_{i=1}^{N} f(x; \xi_i)$ where $x \in \mathbb{R}^d$ is the model parameter and $\{\xi_i\}_{i \in [N]}$ represents training samples. One of the most popular zeroth-order algorithms relies on two function evaluations in opposite directions to estimate the gradients. Such a two-point estimator takes the updating rule $G(x, \rho, u) := (1/2\rho)[f(x+\rho u) - f(x-\rho u)]u$, where $u$ is a random direction sampled from some distribution $\mu(u)$ (e.g., uniformly from the sphere $\mathcal{U}(\sqrt{d}\mathbb{S}^{d-1})$ or Gaussian $\mathcal{N}(0, I_d)$), and $\rho > 0$ is a smoothing parameter.

Under mild assumptions, it is known that the two-point estimator optimizes an approximated, smooth version of the original function, i.e., $\mathbb{E}_u[G(x, \rho, u)] = \nabla_x \mathbb{E}_\epsilon[f(x+\rho\epsilon)]$[1] (Flaxman et al., 2004). In other words, the zeroth-order method is effectively minimizing the expected loss $\mathbb{E}_\epsilon[f(x+\rho\epsilon)]$ in some perturbed neighborhood around $x$. Under such interpretation, zeroth-order optimization has a critical benefit that it is not originally designed for—ensuring the loss is small on average within the neighborhood so that the local minima can be flatter (Wen et al., 2022; Tahmasebi et al., 2024; Zhang et al., 2025). Such a connection has also been studied by prior work (Zhang et al., 2025).

When the model is over-parameterized and non-convex, optimizing a sharpness-aware objective is an effective technique that improves generalization performance (Foret et al., 2020; Bahri et al., 2021). However, the aforementioned vanilla zeroth-order estimate can only achieve a special average-loss based SAM objective, as opposed to the canonical min-max SAM formulation (Foret et al., 2020) or other variants, which have been extensively studied in prior work and demonstrated strong empirical performance (Wu et al., 2020; Sherborne et al., 2023).

---

[1] The distribution of $\epsilon$ depends on that of $u$; see Section 3 for details.

In this work, observing the connections between zeroth-order optimization and SAM, we explore in detail another dimension of zeroth-order optimization, focusing on its explicit bias towards flat solutions. We develop new zeroth-order algorithms that solve a continuous spectrum of sharpness-aware objectives, ranging from the average-loss-based one to the min-max formulation, leveraging exponential tilting. Exponential tilting has been used as a common technique to create parametric distribution shifts in various contexts (Dembo, 2009; Li et al., 2023; Robey et al., 2022), thus providing a smooth transition from min-avg to min-max optimization. It has also been used to develop new sharpness-aware objectives that reweight different local minima (Li et al., 2024). Our zeroth-order algorithms solve a similar objective to arrive at more flat solutions, while preserving the same computational and memory efficiency as the classic zeroth-order estimators.

To be more specific, we consider a soft SAM objective parameterized by a tilting parameter $t$ (named tilted SAM (Li et al., 2024)) that covers the average and min-max formulation as special cases. To approximate the gradient of the tilted SAM objective, we propose different strategies based on finite function evaluations under random perturbations of the model parameter. Additionally, we provide the precise characterizations of a family of sharpness notions of the tilted SAM framework and the solutions it favors, as a function of the tilting parameter $t$. While our framework in principle applies to any form of perturbations, we investigate the cases with Gaussian and ball-constrained uniform perturbation distributions in detail.

Our zeroth-order exponential-tilted sharpness-aware training (ZEST) approach achieves superior performance compared with vanilla two-point estimator (corresponding to solving an average-loss based sharpness-aware objective) across various model types and downstream tasks, including classification, multiple choice QA, and language generation (Section 5). In applications where zeroth-order optimization is competitive in general, ZEST can even achieve higher accuracies than first-order SAM variants while being gradient-free and memory-efficient (Section 5.2).

In summary, our contributions are as follows. In Section 3, we propose a new zeroth-order optimization algorithm (ZEST) that uses exponential tilting to recover a smooth spectrum of sharpness-aware objectives. **Theoretically**, we analyze the explicit bias (the "sharpness" notion) of the exponentially tilted objective and illustrate how ZEST can reach flatter minima than the baselines. We show that our method can identify and conservatively avoid minima with large curvatures in any direction, while vanilla methods cannot (Section 4). **Empirically**, in Section 5, we evaluate ZEST on comprehensive language tasks and different model types, demonstrating that ZEST performs better than MeZO while being equally fast and memory-efficient.

## 2 PRELIMINARIES AND RELATED WORK

**Zeroth-Order Optimization.** Zeroth-order methods have gained recent attention due to their promising performance in fine-tuning language models and their memory efficiency, at the cost of increased iteration complexity compared to first-order methods (Malladi et al., 2023; Zhang et al., 2023; 2024; Gautam et al., 2024). Zeroth-order methods typically optimize for a smoothed version of the functions, which can be interpreted as the expected loss values under perturbed model parameters. Enforcing that the loss values are small in expectation has connections with a special case of sharpness-aware approaches (Zhang et al., 2025), where the sharpness is defined as the trace of the Hessian of gradients (Wen et al., 2022). In this work, we develop new zeroth-order algorithms that solve a spectrum of SAM objectives that cover this special case (Section 3), and provide precise characterizations of the sharpness in our approach (Section 4).

**Sharpness-Aware Minimization.** Sharpness-Aware Minimization (SAM) and its variants have been extensively studied in prior work (Foret et al., 2020; Liu et al., 2022; Kwon et al., 2021; Bartlett et al., 2023; Mi et al., 2022; Ye et al., 2024; Du et al., 2021; Wen et al., 2022; Baek et al., 2024; Tahmasebi et al., 2024; Andriushchenko & Flammarion, 2022; Long & Bartlett, 2024). The canonical SAM objective is to minimize the worst-case loss over perturbed parameters so that the loss values are uniformly small near the local minimum (Foret et al., 2020). The problem is defined as

$$\min_x \max_{\|\epsilon\| \le \rho} f(x+\epsilon), \tag{1}$$

where $\rho$ is the radius of the ball around $x$. To fully realize the potential of zeroth-order approaches and due to the difficulty in optimizing for this objective without gradient access, we propose to

leverage an exponentially-tilted objective that can smoothly approximate this min-max formulation. In particular, we leverage the Tilted Sharpness-Aware Minimization ($t$-SAM) objective (Li et al., 2024), which is paramaterized by a hyperparameter $t>0$ as

$$F_t(x) = \frac{1}{t}\log\mathbb{E}_{\mu(\epsilon)}\Big[e^{tf(x+\epsilon)}\Big], \tag{2}$$

where $\mu(\cdot)$ denotes the distribution density of the perturbation. For instance, $\mu(\epsilon)$ can be the uniform distribution over an $L_2$ ball with radius $\rho$, i.e., $\|\epsilon\|\le\rho$. This objective has been shown to outperform the vanilla SAM formulation Eq. (1) for $0<t<\infty$. When $t\to0$, we have $F_t(x)\to\mathbb{E}_{\|\epsilon\|\le\rho}[f(x+\epsilon)]$, and optimizing it effectively corresponds to running gradient descent using vanilla zeroth-order gradient estimators. As $t\to\infty$, $F_t(x)\to\max_{\|\epsilon\|\le\rho}f(x+\epsilon)$. We note that although ZEST covers a family of sharpness-regularized TSAM objectives, there exist other sharpness-aware objectives and sharpness definitions that we leave for future work (Tahmasebi et al., 2024; Ye et al., 2024).

## 3 Zeroth-Order Tilted Sharpness-Aware Learning

In this section, we introduce our zeroth-order gradient estimates for the $t$-SAM objective (Eq. (2)), so that various $t\in(0,\infty)$ impose different geometry properties of the loss landscape near $x$. In Section 3.1, we first derive the zeroth-order gradient estimate that is equal to the first-order gradient formula of $t$-SAM when the expectation terms (over both the uniform ball and Gaussian distribution) are exactly computed. Next, in Section 3.2, we use two popular ratio estimates to compute the gradient when we can only sample a small finite number of perturbations in practice. Our complete algorithm is presented in Algorithm 1.

### 3.1 Tilted Zeroth-Order Gradient

In this section, we formally present the zeroth-order gradient for the tilted objective. Intuitively, since the gradient of the tilted objective is $\nabla_x F_t(x) = \frac{\mathbb{E}_{\mu(\epsilon)}[e^{tf(x+\epsilon)}\nabla f(x+\epsilon)]}{\mathbb{E}_{\mu(\epsilon)}[e^{tf(x+\epsilon)}]}$, its gradient-free equivalence requires mechanisms that substitute the integration of gradients with the integration of function values. To achieve this, we leverage the divergence theorem (Munkres, 2018) for the case of uniform ball perturbation and Stein's lemma (Chen et al., 2010) for Gaussian perturbations. Under mild assumptions, we have the following theorem to approximate $t$-SAM gradients.

**Theorem 3.1** (Tilted Zeroth-Order Gradient)**.** *Denote $\mathcal{N}:=\mathcal{N}(0,I_d)$, $\mathcal{S}:=\mathcal{U}(\sqrt{d}\mathbb{S}^{d-1})$, and $\mathcal{B}:= \mathcal{U}(\sqrt{d}\mathbb{B}^d)$. Let $\rho$ be a perturbation scale, and let $f(x)<\infty$ and $t\in(0,\infty)$ be such that $\int_v e^{tf(x+\rho v)}dv$ is integrable for any $x$ in the optimization trajectory with $v$ sampled from $\mathcal{N}$ or $\mathcal{B}$. Then the $t$-SAM objective (2) has* exact *zeroth-order gradients. Specifically,*

*(1) with $F_t(x)=\frac{1}{t}\log\mathbb{E}_{v\sim\mathcal{N}}[e^{tf(x+\rho v)}]$, we have*

$$\nabla_x F_t(x) = \frac{1}{t\rho}\frac{\mathbb{E}_{v\sim\mathcal{N}}[(e^{tf(x+\rho v)}-e^{tf(x-\rho v)})v]}{\mathbb{E}_{v\sim\mathcal{N}}[e^{tf(x+\rho v)}+e^{tf(x-\rho v)}]}; \tag{3}$$

*(2) with $F_t(x)=\frac{1}{t}\log\mathbb{E}_{v\sim\mathcal{B}}[e^{tf(x+\rho v)}]$, we have*

$$\nabla_x F_t(x) = \frac{1}{t\rho}\frac{\mathbb{E}_{v\sim\mathcal{S}}[(e^{tf(x+\rho v)}-e^{tf(x-\rho v)})v]}{\mathbb{E}_{v\sim\mathcal{B}}[e^{tf(x+\rho v)}+e^{tf(x-\rho v)}]} \tag{4}$$

$$\approx \frac{1}{t\rho}\frac{\mathbb{E}_{v\sim\mathcal{S}}[(e^{tf(x+\rho v)}-e^{tf(x-\rho v)})v]}{\mathbb{E}_{v\sim\mathcal{S}}[e^{tf(x+\rho v)}+e^{tf(x-\rho v)}]}. \tag{5}$$

*Additionally, assume that $f(x;\xi)$ is $L$-smooth and $M$-Lipschitz for any $x,\xi$. Then with choosing $\rho\le d^{-\frac{4}{5}}$, the bias between Eq. (4) and (5) is controlled by*

$$\|Bias(x)\| = \|(4)-(5)\| \le O\Big(\frac{1}{\sqrt{d}}\Big).$$

We present the proofs in Appendix B and make two remarks here. First, as $t\to0$, $t$-SAM reduces to the average-loss SAM objective $\mathbb{E}[f(x+\epsilon)]$, and our tilted zeroth-order gradient also reduces to the

vanilla zeroth-order gradient. As $t\to\infty$, $t$-SAM approaches the max-loss SAM objective (Eq. (1)), while Theorem 3.1 approaches the regime where integrability fails and thus does not hold — It is expected because max-loss SAM is not differentiable. Second, we use Eq. (5) to approximate (4) to reuse the sampled perturbations on the sphere to compute the denominator. Theoretically, the bias of this approximation reduces by $O(1/\sqrt{d})$ since most of the volume of a high-dimensional ball is concentrated near its boundary (the sphere).

## 3.2 ESTIMATES OF RATIO-OF-EXPECTATIONS

The tilted zeroth-order gradients (Eq. (3) and (5)) compute the ratio of expectations w.r.t. the sampled perturbations. In practice, we only sample a finite number of perturbations and compute the loss average (Malladi et al., 2023; Zhang et al., 2023; Tang et al., 2024). There are multiple well-studied ratio estimates in the Statistics literature (Tin, 1965). In this section, we derive two economic choices, discuss their bias, and propose our ZEST algorithm that leverages finite-perturbation estimates.

Note that in each iteration, we sample $k$ perturbations and denote $a_i^+ = e^{tf(x+\rho v_i)}$, $a_i^- = e^{tf(x-\rho v_i)}$, $Z = \sum_{i=1}^{k} a_i^+ + a_i^-$. We aim to estimate $\frac{\mathbb{E}[A]}{\mathbb{E}[B]}$ with samples $A_i = (a_i^+ - a_i^-)v_i$ and $B_i = a_i^+ + a_i^-, i \in [k]$.

**Naive Plug-In.** A natural ratio estimate is $\frac{\bar{A}}{\bar{B}}$ where $\bar{A}$ and $\bar{B}$ are the sample means for the current iteration. Therefore, we sample $\{v_i\}_{i\in[k]}$ from the given perturbation distribution and compute the sample mean of the numerator and denominator, respectively, which gives us

$$G^k := \frac{1}{t\rho} \frac{\sum_{i=1}^{k}(a_i^+ - a_i^-)v_i}{\sum_{i=1}^{k}(a_i^+ + a_i^-)} = \frac{1}{t\rho}\sum_{i=1}^{k}\frac{a_i^+ - a_i^-}{Z}v_i. \tag{6}$$

Note that due to $\mathbb{E}[\frac{\bar{A}}{\bar{B}}] \neq \frac{\mathbb{E}[A]}{\mathbb{E}[B]}$ by Jensen's inequality, the naive plug-in is only asymptotically unbiased. When $k < \infty$, its bias reduces at rate $O(1/k)$ (Ogliore et al., 2011).

**Bias-Corrected Plug-In.** Due to the small $k$ used in practice, we apply Taylor expansion to derive the bias-corrected estimate, following Van Kempen & Van Vliet (2000b). With the normalized values $\bar{a}_i^+ := a_i^+/Z$ and $\bar{a}_i^- := a_i^-/Z$, we have

$$G_{\text{BC}}^k := \frac{1}{t\rho}\sum_{i=1}^{k}\left\{1 + \frac{k}{k-1}[\bar{a}_i^+ + \bar{a}_i^- - \sum_{i=1}^{k}(\bar{a}_i^+ + \bar{a}_i^-)^2]\right\}(\bar{a}_i^+ - \bar{a}_i^-)v_i, \tag{7}$$

and the derivation is in Appendix B.4. $G_{\text{BC}}^k$ has an improved bias reduction rate $O(1/k^2)$ (Van Kempen & Van Vliet, 2000a) and has the same memory/computational complexity as $G^k$ and the vanilla zeroth-order gradient estimator (since the computation and storage of the exponential of $k$ loss values is negligible). We provide a synthetic example in Figure 3 to demonstrate the reduced bias of $G_{\text{BC}}^k$.

With the above two options derived, we present our ZEST algorithm and present its memory-efficient implementation in Algorithm 1. In each iteration, we first sample $k$ perturbations iteratively using random seeds and record the normalized tilted loss values (Line 3-7). For memory efficiency, the perturbations will be deleted once these loss values are computed. Next, we obtain the weight for each perturbation using the chosen ratio estimate (Line 8-9). Finally, we re-generate the perturbations via the same random seeds and update the model parameters (Line 10-13). Since we sample and recover the perturbations in place without storing them in memory, ZEST is more memory-efficient than the first-order optimizer for $t$-SAM (Li et al., 2024). See a detailed memory analysis in Section 5.

## 4 THE NOTION OF SHARPNESS

In this section, we analyze the explicit bias (i.e., sharpness notions) of the $t$-SAM objective under both Gaussian (Section 4.1) and uniform ball perturbation (Section 4.2). Recall that updating via the vanilla zeroth-order gradient estimator is essentially minimizing $\mathbb{E}_v[f(x+\rho v)]$, which can be decomposed to the empirical loss $f(x)$ term plus a sharpness regularization term $R_{\text{avg}} \propto \text{Tr}(\nabla^2 f(x))$ (Section 2). We decompose the $t$-SAM objective into

$$F_t(x) = f(x) + R_t(x) + O(\rho^2 d)$$

---

**Algorithm 1:** ZEST

**Input :** $x\in\mathbb{R}^d$, tilting parameter $t$, perturbation scale $\rho$, number of queries $k$, learning rate $\eta$

1  **for** each iteration **do**
2      Sample a batch of training data $\mathcal{D}$ and seeds $\{s_i\}_{i\in[k]}$
3      **for** $i=1,\cdots,k$ **do**
4          Sample $v_i\sim\mathcal{N}(0,I_d)$ or $\mathcal{U}(\sqrt{d}\mathbb{S}^{d-1})$ based on seed $s_i$
5          Compute $a_i^+\leftarrow e^{tf(x+\rho v_i;\mathcal{D})}$, $a_i^-\leftarrow e^{tf(x-\rho v_i;\mathcal{D})}$
6      **end**
7      Compute $Z\leftarrow\sum_{i=1}^k a_i^+ + a_i^-$ and $\bar{a}_i^+\leftarrow a_i^+/Z$, $\bar{a}_i^-\leftarrow a_i^-/Z$ for $i\in[k]$
8      Compute $w_i$ for $i\in[k]$ by
9          **Option 1 (Naive):** $w_i\leftarrow\bar{a}_i^+ - \bar{a}_i^-$

           **Option 2 (Bias-corrected):** $w_i\leftarrow\left\{1+\frac{k}{k-1}[\bar{a}_i^+ + \bar{a}_i^- - \sum_{i=1}^k (\bar{a}_i^+ + \bar{a}_i^-)^2]\right\}(\bar{a}_i^+ - \bar{a}_i^-)$

10      **for** $i=1,\cdots,k$ **do**
11          Recover $v_i\sim\mathcal{N}(0,I_d)$ or $\mathcal{U}(\sqrt{d}\mathbb{S}^{d-1})$ based on seed $s_i$
12          $x\leftarrow x-\eta(w_i/t\rho)*v_i$
13      **end**
14 **end**

---

where $f(x)$ is the empirical loss, $R_t(x)$ is the regularizer (used as our *sharpness* notion) dependent on $t$, and $O(\rho^2 d)$ is the Taylor expansion error that can be controlled by taking proper $\rho$'s. Across two perturbation distributions, we show that as $t\to 0$, $R_t$ reduces to $R_{\text{avg}}$; as $t$ increases, $R_t$ increasingly relies on the gradient component in the top eigenspace of the Hessian $\nabla^2 f(x)$ and its top eigenvalues; as $t\to\infty$ (when admissible), $R_t$ exclusively relies on the gradient component projected to the first Hessian eigenvector and the largest eigenvalue. Therefore, our regularizer $R_t$ represents a spectrum of sharpness notions that promote "flatter" solutions. In Section 4.3, we present a low-dimensional toy problem to illustrate (1) the different convergence behaviors of ZEST in contrast to vanilla zeroth-order methods due to different sharpness notions and (2) when and how our notion is superior.

In the following, we start by defining *sharpness sensitivity*, a notion that describes how the value of an eigenvalue impacts the sharpness regularizer $R_t$.

**Definition 4.1** (Sharpness Sensitivity). *The dependence of sharpness $R_t$ on $x$ can be re-expressed as its dependence on the Hessian eigenvalues $\{\lambda_i\}_{i=1}^d$ and the components of $\nabla f(x)$ in the eigenspace. We define the sharpness sensitivity to an arbitrary $\lambda_i$ as*

$$\phi_i(t):=\frac{\partial R_t}{\partial \lambda_i}, \tag{8}$$

*which indicates how much impact the value of an arbitrary $\lambda_i$ has on the value of $R_t$.*

We note that if $\phi_i$ increases as $\lambda_i$ increases, $R_t$ is more dominated by large eigenvalues. Alternatively, if $\phi_i$ is constant regardless of the value of $\lambda_i$, $R_t$ penalizes each eigenvalue equally and thus introduces the bias for solutions with small average eigenvalues. With this quantity, we analyze $R_t$ when the perturbation is sampled from $\mathcal{N}(0,I_d)$ (denoted as $\mathcal{N}$) and $\mathcal{U}(\sqrt{d}\mathbb{B}^d)$ (denoted as $\mathcal{B}$) as follows.

## 4.1   Gaussian Perturbation

The Taylor expansion of $f(x+\rho v)$ with $v\sim\mathcal{N}$ is

$$f(x+\rho v)=f(x)+\rho\nabla f(x)^\top v+\frac{\rho^2}{2}v^\top\nabla^2 f(x)v+O(\rho^2\|v\|^2),$$

where $O(\rho^2\|v\|^2){=}O(\rho^2 d)$ with high probability for large $d$. Therefore, the Taylor expansion of the $t$-SAM objective is

$$F_t(x){=}f(x){+}\underbrace{\frac{1}{t}\log\mathbb{E}_{v\sim\mathcal{N}}\left[\exp\left(t[\rho\nabla f(x)^\top v{+}\frac{\rho^2}{2}v^\top\nabla^2 f(x)v]\right)\right]}_{\text{Regularizer } R_t}{+}O(\rho^2 d). \qquad (9)$$

We decompose Hessian $\nabla^2 f(x)$ into $\nabla^2 f(x){=}Q^\top\Lambda Q$, where $Q$ is orthogonal with columns $\{e_1,...,e_d\}$ that are ordered Hessian eigenvectors, and $\Lambda{=}\text{diag}(\lambda_1,\lambda_2,...,\lambda_d)$ where $\lambda_1{\geq}...{\geq}\lambda_d$ are the ordered Hessian eigenvalues. Denote $g{:=}Q\nabla f(x)$ and thus $g_i$ is the component of the gradient along the $i$-th eigenvector. Then we have the following theorem for $R_t$ with proof in Appendix B.5.

**Theorem 4.1** (Sharpness under Gaussian Perturbation). *Under Gaussian perturbation, if we choose $\rho$ such that $1{-}t\rho^2\lambda_i{>}0$ holds for any $i$, then we have*

$$R_t{=}\frac{1}{2t}\sum_{i=1}^{d}\left[\frac{(t\rho g_i)^2}{1{-}t\rho^2\lambda_i}{-}\log(1{-}t\rho^2\lambda_i)\right]. \qquad (10)$$

We see that as $t{\to}0$, we have $\lim_{t\to 0}R_t{=}\frac{\rho^2}{2}\sum_{i=1}^{d}\lambda_i{=}R_{\text{avg}}$, which is consistent with existing work (Wen et al., 2022; Tahmasebi et al., 2024). As $t$ increases, the regularizer sensitivity $\phi_i(t)$ satisfies

$$\phi_i(t){=}\frac{\rho^2}{2(1{-}t\rho^2\lambda_i)}\left(\frac{t^2\rho^2 g_i^2}{1{-}t\rho^2 g_i}{+}1\right){>}0 \text{ for valid } \rho.$$

It shows that the sensitivity of $R_t$ to $\lambda_i$ is dependent on $t$. When $t{=}0$, the sensitivity is the constant $\rho^2/2$, that is, each eigenvalue contributes the same to $R_t$. As $t$ increases, the sensitivity increases, i.e., $R_t$ will be more dominated by large eigenvalues.

## 4.2 Ball Perturbation

Apart from the Gaussian perturbation, we analyze the regularizer and explicit bias when the ball perturbation is used in this section. The Taylor expansion error of $f(x{+}\rho v)$ with $v\sim\mathcal{U}(\sqrt{d}\mathbb{B}^d)$ is $O(\rho^2 d)$ due to $\|v\|^2{\leq}d$. Therefore, the Taylor expansion of $t$-SAM gives

$$F_t(x){=}f(x){+}\underbrace{\frac{1}{t}\log\mathbb{E}_{\mathcal{U}(\sqrt{d}\mathbb{B}^d)}\left[\exp\left(t[\rho\nabla f(x)^\top v{+}\frac{\rho^2}{2}v^\top\nabla^2 f(x)v]\right)\right]}_{\text{Regularizer } R_t}{+}O(\rho^2 d).$$

and we have the following theorem for $R_t$, whose complete derivation is in Appendix B.6.

**Theorem 4.2** (Sharpness under Ball Perturbation). *Assume that $\|\nabla f(x)\|{<}\infty$ and $\nabla^2 f(x)$ has bounded eigenvalues for any $x$ in our optimization trajectory. Under ball perturbation, $R_t$ is continuous and non-decreasing in $t$ for any $t{<}\infty$, and the regularizer sensitivity $\phi_i$ is continuous and non-decreasing in $\lambda_i$. Therefore, we analyze two extreme cases. When $t{\to}0$,*

$$\lim_{t\to 0}R_t{=}\frac{\rho^2 d}{2(d{+}2)}\sum_{i=1}^{d}\lambda_i, \qquad (11)$$

*which recovers the sharpness of the vanilla zeroth-order methods $R_{avg}$, i.e., a simple average of eigenvalues. When $t{\to}\infty$, we have*

$$\lim_{t\to\infty}R_t{:=}R_\infty{=}\max_{\|u\|\leq\sqrt{d}}\underbrace{\rho g^\top u}_{\text{Slope penalty}}{+}\underbrace{\frac{\rho^2}{2}u^\top\Lambda u}_{\text{Curve penalty}}. \qquad (12)$$

Theorem 4.2 indicates that $R_t$ *pessimistically* regularizes the objective $f(x)$ so that we favor the *flat* solution $\hat{x}$ where $f(\hat{x})$ has neither highly curved directions nor large slopes along the curved directions. We discuss the penalties specifically in three regimes.

**Linear regime.** If $f(x)$ is piecewise-linear within the search space for the next iteration $x'$, the curve penalty is zero and $R_\infty$ depends solely on $\max\{g^\top u:\|u\|{\leq}\sqrt{d}\}{=}\sqrt{d}\|g\|{=}\sqrt{d}\|\nabla f(x)\|$ with $u^\star{=}\sqrt{d}g/\|g\|$. Therefore, $t$-SAM biases against the next iterations with steep slopes (gradients).

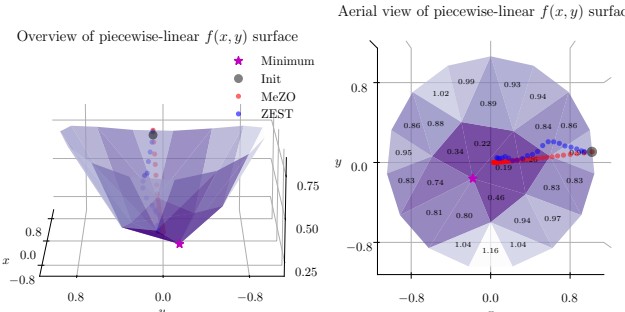
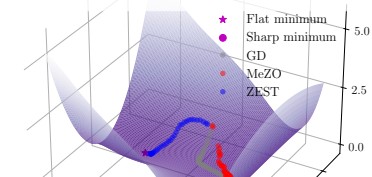

(a) Convergence of different methods that prefer trajectories with different slopes. MeZO does not have a slope regularizer while ZEST identifies flat next iterations with smaller slopes (gradients). The color and the value of each triangle indicate its slope, with *darker* indicating *flatter*.

(b) Convergence of different methods when two minima have the same loss value and average eigenvalues. GD and MeZO converge to the sharp minimum (with larger $\lambda_{\max}$); ZEST converges to the flat one (smaller $\lambda_{\max}$).

Figure 1: Convergence behaviors of different methods on examples for the (a) linear and (b) stationary regimes. It shows that (1) MeZO can make steep steps while ZEST identifies flat next iterations and (2) MeZO can converge to minima with large $\lambda_{\max}$ while ZEST explicitly biases against large $\lambda_{\max}$.

**Stationary regime.** If $f(x)$ has multiple local minima as candidates for the next iteration, the slope penalties for them are all zero and $R_\infty$ depends only on $\max\{u^\top \Lambda u : \|u\| \le \sqrt{d}\} = \sqrt{d}\max(\lambda_1, 0)$ with $u^\star = \sqrt{d}e_1$. Therefore, $F_t$ biases against next iterations with large curvature in *any* direction.

**General case.** When both the curve and slope penalties are active, we use KKT conditions to solve Eq. (12) in Appendix B.7. We have that when $\nabla^2 f(x) \npreceq 0$,

1. Gradient–curvature co-alignment is what matters. Only eigen-directions with nonzero gradient projection ($g_i \ne 0$) influence $R_t$, and the influence grows with both $|g_i|$ and $\lambda_i$.

2. The largest positive eigenvalues dominate if the gradient points there: When $g$ has projections on the top-eigenvector(s), those eigenvalues have the largest impact on $R_t$.

We make two comments based on the above results. First, when $t \to 0$, our regularizer $R_t$ recovers $R_{\text{avg}}$ of the average-loss SAM objective under both Gaussian and uniform ball perturbations. As $t$ increases, regularizer sensitivity increases and thus the penalty from each eigenvalue changes from uniformity to dominance by $\lambda_{\max}$. Second, as $t \to \infty$ under ball perturbation (where the max-loss SAM objective is well-defined), our regularizer is consistent with existing work (Wen et al., 2022) for the general case and the work of Tahmasebi et al. (2024) for the stationary regime.

Furthermore, we discuss how hyperparameters such as $\rho$ and $d$ influence the effective choices of $t$ in Appendix B.8. In the following section, we present two low-dimensional examples that correspond to the linear and stationary regimes, respectively, to illustrate the effects of different biases introduced by $R_t$ in contrast to $R_{\text{avg}}$.

### 4.3 LOW-DIMENSIONAL EXAMPLES

We illustrate the benefit of ZEST's sharpness notion through 2D examples for the linear and stationary regimes. For the linear regime, we create a piecewise-linear loss function with one minimum (Figure 1a). There are multiple routes to reach the minimum, some steep (with large slopes/gradients, lightly colored) and some flat (with small slopes, darkly colored). We observe that though both ZEST and the vanilla zeroth-order algorithm (MeZO (Malladi et al., 2023)) approaches the minimum, ZEST identifies and chooses the flatter route (with darker planes) while MeZO chooses the steep trajectory.

For the stationary regime, we present the $f$ with two local minima, $(\pm 1, 0)$ such that $f(\pm 1, 0) = 0$ (Figure 1b). Denoting the Hessian of $f$ at point $(x, y)$ as $H(x, y)$, we have the eigenvalues of $H(1, 0)$ as $\{\frac{12}{5}, \frac{2}{5}\}$ and those of $H(-1, 0)$ as $\{\frac{10}{5}, \frac{4}{5}\}$. Since the two minima have the same average of eigenvalues (trace), optimizers with sharpness defined as $R_{\text{avg}}$, such as MeZO, would treat these

minima equally sharp. However, the fact that $\lambda_{\max}[H(1,0)] > \lambda_{\max}[H(-1,0)]$ indicates that there exist perturbation directions that substantially impair model utility if it reaches $(1,0)$, which should be avoided in critical applications. Noticeably, we observe that ZEST can avoid the pitfall of $(1,0)$ and arrive at $(-1,0)$ despite having the same loss value and Hessian trace. By taking risk-averse steps, ZEST is sensitive to $\lambda_{\max}$, which is consistent with our analysis.

## 5 EXPERIMENTS

We conduct experiments on masked LMs (RoBERTa-base, 135M (Liu et al., 2019)) on GLUE classification tasks in Section 5.1 and autoregressive LMs (OPT-1.3B (Zhang et al., 2022) and LLaMA-7B (Touvron et al., 2023)) on multiple choice, generation, and classification tasks in Section 5.2. We focus on many-shot settings with prompts, following prior zeroth-order optimization literature (Malladi et al., 2023; Zhang et al., 2023; Chen et al., 2017). Furthermore, we experiment on vision tasks with clean and noisy training labels as in prior sharpness-aware literature (Baek et al., 2024; Li et al., 2024; Foret et al., 2020), with results in Table 6. Across diverse tasks and model types, ZEST is a computational- and memory-efficient alternative to first-order approaches and outperforms the vanilla zeroth-order baseline MeZO (Malladi et al., 2023). In Section 5.3, we evaluate the "flatness" of ZEST solutions under multiple definitions. In Section 5.4, we discuss the effects of the tilting hyperparameter $t$ and practical tuning guidance.

**Baselines.** For each dataset, we perform full-parameter fine-tuning to minimize the ERM objective $f(x)$, average-loss SAM $\mathbb{E}[f(x+\epsilon)]$, max-loss SAM $\max_\epsilon f(x+\epsilon)$, and $t$-SAM $F_t(x)$ via both first-order and zeroth-order methods. We summarize the objectives and memory complexities in Table 1. We present the performance of both $\text{ZEST}_N$ (Option 1) and $\text{ZEST}_{BC}$ (Option 2), which use different update rule options in Algorithm 1 Line 9. Additionally, we highlight that ZEST has the same computational and memory complexity as MeZO since the cost of taking the exponential of a few losses is negligible. Therefore, the empirical memory efficiency and wallclock time measures in MeZO apply to ZEST (Appendix E.7, F.5, and F.6 of Malladi et al. (2023)).

Table 1: Objective and memory cost of different methods. We follow the memory analysis approach in Chen et al. (2017), where $l$ is the layer index, and $a_l$ denotes the stored activations for computing the backward gradients for layer $l$, and $|\cdot|$ denotes the dimension of the vector. We present the memory usage under ball perturbation since it is more costly than sampling from Gaussian.

| Type | Objective | Method | Memory |
|---|---|---|---|
| 1st-order | $f(x)$ | SGD | $\sum_l \max(|a_l|,|x_l|)+|x|$ |
| | $\mathbb{E}[f(x+\epsilon)]$ | ESAM | $\sum_l \max(|a_l|,|x_l|)+2|x|$ |
| | $\max_\epsilon f(x+\epsilon)$ | SAM | $\sum_l \max(|a_l|,|x_l|)+2|x|$ |
| | $F_t(x)$ | TSAM | $\sum_l \max(|a_l|,|x_l|)+(k+1)|x|$ |
| 0th-order | $\mathbb{E}[f(x+\epsilon)]$ | MeZO | |
| | $F_t(x)$ | $\text{ZEST}_N$ | $2|x|$ |
| | | $\text{ZEST}_{BC}$ | |

### 5.1 MASKED LANGUAGE MODELS

We experiment on four types of classification tasks in the GLUE benchmark (Wang et al., 2018), including sentiment classification, paraphrasing, topic classification, and natural language inference. Following prior work (Malladi et al., 2023; Zhang et al., 2023; Chen et al., 2017), we focus on the setting of many-shot fine-tuning with prompts where we sample 512 samples for each class. Since prior work shows that SAM is robust to label noise (Baek et al., 2024; Li et al., 2024), we additionally fine-tune on the noisy version of each dataset where the label noises are created by switching 30% of the true labels uniformly at random to other labels (details in Appendix C).

On clean data, ZEST consistently outperforms MeZO by 0.1% to 1.7% in accuracy (Table 2), and on data with noisy labels, ZEST consistently outperforms MeZO by 0.4% to 5.9% in accuracy (Table 3). On clean and noisy data, $\text{ZEST}_{BC}$ outperforms $\text{ZEST}_N$ on 3/8 and 4/8 tasks, respectively.

Table 2: Experiments on RoBERTa-Base (512 training examples per class).

| Type | Task
Task type | SST-2 | SST-5
sentiment cls. | QQP | MRPC
paraphrase | TREC
topic cls. | MNLI | SNLI
natural language inference | RTE |
|---|---|---|---|---|---|---|---|---|---|
| 1st-
order | SGD | 92.8 | 56.2 | 84.0 | 88.2 | 97.6 | 78.4 | 84.7 | 78.3 |
| | ESAM | 93.0 | 56.4 | 84.3 | 88.5 | 97.8 | 78.4 | 85.3 | 79.4 |
| | SAM | 93.2 | 56.4 | 84.8 | **90.0** | 97.8 | 79.3 | 85.4 | 80.1 |
| | TSAM | **93.5** | **57.5** | **85.0** | 89.2 | **98.0** | **79.5** | **85.8** | **80.5** |
| 0th-
order | MeZO | 92.1 | 48.6 | 71.4 | 81.9 | 94.8 | 71.8 | 78.2 | 72.9 |
| | ZEST$_N$ | **92.2** | 49.4 | 71.6 | **83.6** | **95.6** | 73.6 | **78.3** | **73.3** |
| | ZEST$_{BC}$ | 92.0 | **49.7** | **72.6** | 81.6 | 95.2 | **73.8** | 78.2 | 72.9 |

Table 3: Experiments on RoBERTa-Base (512 training examples per class with 30% noisy labels).

| Type | Task
Task type | SST-2 | SST-5
sentiment cls. | QQP | MRPC
paraphrase | TREC
topic cls. | MNLI | SNLI
natural language inference | RTE |
|---|---|---|---|---|---|---|---|---|---|
| 1st-
order | SGD | 89.2 | 53.7 | 73.8 | 77.0 | 96.2 | 73.8 | 78.1 | 66.1 |
| | ESAM | 89.9 | 54.6 | 79.5 | 77.5 | 96.2 | 75.4 | 79.2 | 66.8 |
| | SAM | 91.1 | **55.2** | 80.2 | **78.9** | 96.2 | **76.9** | 80.8 | **68.6** |
| | TSAM | **91.5** | **55.2** | **81.0** | 77.7 | **96.4** | 76.5 | **81.4** | 67.5 |
| 0th-
order | MeZO | 89.0 | 44.7 | 62.4 | 67.2 | 86.2 | 60.3 | 59.2 | 59.9 |
| | ZEST$_N$ | **89.4** | **46.2** | **68.3** | 68.6 | **86.8** | **63.4** | **64.9** | 61.4 |
| | ZEST$_{BC}$ | 88.2 | 44.7 | 62.7 | **68.9** | **86.8** | **63.4** | 64.3 | **61.7** |

## 5.2 AUTOREGRESSIVE LANGUAGE MODELS

Furthermore, we experiment on multiple-choice, generation, and classification tasks with OPT-1.3B and LLaMA-7B. For each dataset, we randomly sample 1000, 500, and 1000 examples for training, validation, and testing. From Table 4, we observe that (1) TSAM and SAM consistently outperform ESAM, indicating the superiority of pessimistic sharpness notion as opposed to $R_{\text{avg}}$; (2) ZEST consistently outperforms MeZO and matches/outperforms first-order methods on multiple tasks.

Table 4: Test accuracies/F1 of OPT-1.3B (1000 training samples).

| Type | Task
Task type | COPA | ReCoRD
multiple choice | SQuAD | DROP
generation | WSC | WIC
classification |
|---|---|---|---|---|---|---|---|
| 1st-
order | SGD | 75.0 | 72.2 | 83.4 | 29.7 | 57.8 | 65.2 |
| | ESAM | 76.0 | 72.5 | 83.7 | 31.2 | 58.8 | 66.8 |
| | SAM | **77.0** | **72.7** | 84.3 | **31.8** | 64.6 | **68.7** |
| | TSAM | **77.0** | 72.1 | **84.6** | 31.3 | **66.5** | 67.1 |
| 0th-
order | MeZO | 74.0 | 72.4 | 78.8 | 25.2 | 61.5 | 55.5 |
| | ZEST$_N$ | **78.0** | 72.3 | **79.4** | 25.5 | **64.4** | 57.1 |
| | ZEST$_{BC}$ | 77.0 | **72.5** | 79.0 | **25.7** | 63.5 | **57.2** |

We observe that ZEST$_{BC}$ and ZEST$_N$ perform on par in the above experiments, which we attribute to variance across trials. In Figure 3, we show that the bias-corrected estimator indeed has smaller bias, but the two estimators have approximately the same variance. In practice, one samples a small number of random seeds, which makes their performance appear similar due to variance. Despite this, we introduce both estimators to illustrate that our method is general and admits various ratio estimators. Additionally, with abundant computing resources to run many trials, it is favorable to leverage bias correction. We leave applying more advanced ratio estimates to ZEST for future work.

Table 5: Test accuracy of LLaMA-7B (1000 training samples).

| Type | Task
Task type | COPA | ReCoRD | WIC | WSC |
|------|------|------|------|------|------|
|      |      | multiple choice | | classification | |
| 1st-
order | SGD | 85.0 | 82.4 | 61.5 | 66.5 |
|      | ESAM | 85.0 | 82.5 | 62.2 | 65.7 |
|      | SAM | **86.0** | **82.8** | **63.5** | **67.9** |
|      | TSAM | OOM | OOM | OOM | OOM |
| 0th-
order | MeZO | **89.0** | 80.1 | 60.8 | 64.4 |
|      | ZEST$_N$ | **89.0** | **81.8** | 62.7 | **66.2** |
|      | ZEST$_{BC}$ | **89.0** | 81.3 | **63.4** | 65.9 |

Table 6: Test accuracy of ViT on CIFAR-10 w. clean/noisy labels.

| Method | Clean | 30% Noisy |
|------|------|------|
| SGD | 96.9 | 95.1 |
| ESAM | 97.5 | 95.7 |
| SAM | **97.9** | **97.5** |
| TSAM | **97.9** | 97.4 |
| MeZO | 80.2 | 69.0 |
| ZEST$_N$ | **82.8** | **72.3** |
| ZEST$_{BC}$ | 82.4 | 71.8 |

## 5.3 FLATNESS OF ZEST SOLUTIONS

In this section, we evaluate the flatness of ZEST solutions in comparison to MeZO solutions by comparing their sharpness measurements under various definitions, including the average loss in the neighborhood of $x$ (Wen et al., 2023) and top-5 eigenvalues of the Hessian (Wen et al., 2022). In Figure 2 (left), we observe that under various neighborhood radii, the minimum found by ZEST has smaller average losses than that found by MeZO. In addition, the top-5 eigenvalues are all smaller than those of MeZO (right). The same observation made on more datasets is presented in Appendix C.2.

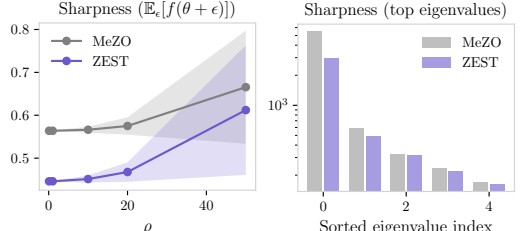

Figure 2: Sharpness of the solutions found by MeZO and ZEST on MRPC. Left: Scatters denote the average loss of the neighborhood among 500 perturbations, and the shade denotes the standard deviation. Right: Top-5 eigenvalues of Hessian.

## 5.4 SENSITIVITY TO $t$

Though the generalization bounds for exponential tilting are presented in prior literature (Li et al., 2024; Aminian et al., 2025), the optimal choice of $t$ is problem-dependent. In practice, one needs to find the $t$ value that yields the best validation performance. In this section, we present the validation performance of RoBERTa-Base under $t=\{0,1,5,20\}$ in Figure 4 in the appendix. The results show that multiple $t$ values yield superior performance to MeZO ($t=0$). Additionally, $t=1$ is a safe go-to choice for preliminary trials since it almost always yields superior performance to MeZO: $t=1$ matches or outperforms MeZO on 7/8 settings; the only case when $t=1$ underperforms is by 0.1%.

## 6 CONCLUSION

We introduce ZEST, a gradient-free optimization framework that unifies classic zeroth-order optimization with sharpness-aware minimization. By leveraging exponential tilting, ZEST optimizes for a continuous spectrum of objectives that smoothly interpolate between the standard average-loss zeroth-order objective and the worst-case min-max SAM formulation. Theoretically, we characterize the sharpness bias induced by the tilted objective and demonstrate that ZEST can avoid minima of high curvatures that vanilla zeroth-order methods overlook. Empirically, ZEST preserves efficiency while consistently outperforming vanilla zeroth-order methods and, in many cases, first-order SAM variants on various downstream tasks. These demonstrate that ZEST provides a powerful bridge between zeroth-order optimization and sharpness-aware training, enabling gradient-free yet curvature-sensitive learning that generalizes better while remaining efficient.

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

## A    VANILLA ZEROTH-ORDER GRADIENT ESTIMATE

In this section, we provide an additional introduction to zeroth-order optimization and the vanilla gradient estimate.

In zeroth-order optimization, we estimate $\nabla f(x)$ using only function evaluations. A standard estimator is the two-point symmetric finite difference

$$G(x,\rho,u):=\frac{f(x+\rho u)-f(x-\rho u)}{2\rho}u, \tag{13}$$

where $u$ is a random direction sampled uniformly from the sphere $\sqrt{d}\mathbb{S}^{d-1}$ or Gaussian $\mathcal{N}(0,I_d)$, and $\rho>0$ is a smoothing parameter. In the following, we abbreviate $\mathcal{B}:=\mathcal{U}(\sqrt{d}\mathbb{B}^d)$, $\mathcal{S}:=\mathcal{U}(\sqrt{d}\mathbb{S}^{d-1})$, and $\mathcal{N}:=\mathcal{N}(0,I_d)$. We use $\mathbb{E}_{\mathcal{B}}$, $\mathbb{E}_{v\sim\mathcal{B}}$, and $\mathbb{E}_{v\sim\mathcal{U}(\sqrt{d}\mathbb{B}^d)}$ interchangeably when the meaning is clear from the context.

For sampling from the sphere, when $\rho\to 0$, the estimator is unbiased since

$$\mathbb{E}_{u\sim\mathcal{S}}\left[\frac{f(x+\rho u)-f(x-\rho u)}{2\rho}u\right]\to\mathbb{E}_{u\sim\mathcal{S}}[uu^\top]\nabla f(x)=\nabla f(x).$$

When $\rho$ is general, the estimator corresponds to the gradient of a smoothed objective (Duchi et al., 2015; Zhang et al., 2023). Define

$$f_\rho(x):=\mathbb{E}_{v\sim\mathcal{B}}[f(x+\rho v)]$$

and by the divergence theorem in $\mathbb{R}^d$,

$$\nabla_x f_\rho(x)=\mathbb{E}_{u\sim\mathcal{S}}[G(x,\rho,u)].$$

Thus, the estimator in expectation is the gradient of a smoothed version of $f$ where the smoother is a uniform distribution on a ball. Similarly, for sampling from Gaussian, we have

$$\nabla_x\mathbb{E}_{v\sim\mathcal{N}}[f(x+\rho v)]=\mathbb{E}_{v\sim\mathcal{N}}[G(x,\rho,v)].$$

We can interpret the above results that updating using the vanilla zeroth-order gradient estimate optimizes for a smoothed objective of $f(x)$. By Taylor expansion, for $\pi\in\{\mathcal{S},\mathcal{N}\}$, we have

$$\mathbb{E}_{v\sim\pi}[f(x+\rho v)]=f(x)+\mathbb{E}_{v\sim\pi}[\nabla f(x)^\top v]+\frac{\rho^2}{2}\mathbb{E}_{v\sim\pi}[v^\top\nabla^2 f(x)v]+\mathbb{E}_{v\sim\pi}[O(\rho^2\|v\|^2)]$$

$$=f(x)+\frac{\rho^2}{2}\mathrm{Tr}(\nabla^2 f(x))+O(\rho^2 d),$$

which implies that the effective objective of vanilla zeroth-order optimization is the empirical loss $f(x)$ added by a regularizer $R_{\mathrm{avg}}\propto\mathrm{Tr}(\nabla^2 f(x))$.

## B    PROOFS

### B.1    PROOF OF THEOREM 3.1 (GAUSSIAN)

*Proof.* By Stein's lemma (Chen et al., 2010), for the $d$-dimensional random vector $v\sim\mathcal{N}(0,I_d)$ and a differentiable function $g$ for which $\mathbb{E}[g(v)v]$ and $\mathbb{E}[\nabla_v g(v)]$ both exist, we have

$$\mathbb{E}_v[g(v)v]=\mathbb{E}_v[\nabla_v g(v)]. \tag{14}$$

Therefore, we let $g(v)=e^{tf(x+\rho v)}$ and obtain

$$\int_v\nabla_v(e^{tf(x+\rho v)})p(v)dv=\int_v e^{tf(x+\rho v)}p(v)v dv$$

and thus

$$\int_v e^{tf(x+\rho v)-\|v\|^2/2}\nabla_v f(x+\rho v)dv=\frac{1}{t}\int_v e^{tf(x+\rho v)-\|v\|^2/2}v dv. \tag{15}$$

Note that the gradient of $t$-SAM is

$$\nabla_x F_t(x) = \frac{\mathbb{E}_{v\sim\mathcal{N}}[e^{tf(x+\rho v)}\nabla f(x+\rho v)]}{\mathbb{E}_{v\sim\mathcal{N}}[e^{tf(x+\rho v)}]} = \frac{\int_v e^{tf(x+\rho v)-\|v\|^2/2}\nabla f(x+\rho v)dv}{\int_v e^{tf(x+\rho v)-\|v\|^2/2}dv}.$$

Combining the above, we have

$$\nabla_x F_t(x) \overset{(a)}{=} \frac{\int_v e^{tf(x+\rho v)-\|v\|^2/2}\nabla_v f(x+\rho v)dv}{\rho\int_v e^{tf(x+\rho v)-\|v\|^2/2}dv}$$

$$\overset{(15)}{=} \frac{\int_v e^{tf(x+\rho v)-\|v\|^2/2}v\,dv}{t\rho\int_v e^{tf(x+\rho v)-\|v\|^2/2}dv}$$

$$= \frac{\mathbb{E}_{v\sim\mathcal{N}}[e^{tf(x+\rho v)}v]}{t\rho\mathbb{E}_{v\sim\mathcal{N}}[e^{tf(x+\rho v)}]} \tag{16}$$

$$= \frac{\frac{1}{2}\left(\mathbb{E}_{v\sim\mathcal{N}}[e^{tf(x+\rho v)}v]+\mathbb{E}_{v\sim\mathcal{N}}[e^{tf(x+\rho v)}v]\right)}{t\rho\cdot\frac{1}{2}\left(\mathbb{E}_{v\sim\mathcal{N}}[e^{tf(x+\rho v)}]+\mathbb{E}_{v\sim\mathcal{N}}[e^{tf(x+\rho v)}]\right)}$$

$$= \frac{1}{t\rho}\frac{\mathbb{E}_{v\sim\mathcal{N}}[e^{tf(x+\rho v)}v]+\mathbb{E}_{v\sim\mathcal{N}}[e^{tf(x+\rho(-v))}(-v)]}{\mathbb{E}_{v\sim\mathcal{N}(0,I_d)}[e^{tf(x+\rho v)}]+\mathbb{E}_{v\sim\mathcal{N}}[e^{tf(x+\rho(-v))}]}$$

$$= \frac{1}{t\rho}\frac{\mathbb{E}_{v\sim\mathcal{N}}[(e^{tf(x+\rho v)}-e^{tf(x-\rho v)})v]}{\mathbb{E}_{v\sim\mathcal{N}}[e^{tf(x+\rho v)}+e^{tf(x-\rho v)}]} \tag{17}$$

where $(a)$ is due to $\nabla_x\phi(x+\rho v)=\nabla\phi(x+\rho v)=\frac{1}{\rho}\nabla_v\phi(x+\rho v)$ where $\nabla\phi(\cdot)$ denotes the gradient w.r.t. the input of function $\phi$.

$\square$

**Case of $t\to0$.** As $t\to0$, we apply L'Hôpital's rule to obtain

$$\lim_{t\to0}\nabla_x F_t(x) = \frac{\lim_{t\to0}\mathbb{E}_{\mathcal{N}}[e^{tf(x+\rho v)}f(x+\rho v)v]}{\lim_{t\to0}\rho\mathbb{E}_{\mathcal{N}}[e^{tf(x+\rho v)}]+t\rho\mathbb{E}_{\mathcal{N}}[e^{tf(x+\rho v)}f(x+\rho v)]}$$

$$= \mathbb{E}_{\mathcal{N}}\left[\frac{f(x+\rho v)v}{\rho}\right]$$

$$= \mathbb{E}_{\mathcal{N}}\left[\frac{f(x+\rho v)v}{2\rho}\right]+\mathbb{E}_{\mathcal{N}}\left[\frac{f(x+\rho(-v))(-v)}{2\rho}\right]$$

$$= \mathbb{E}_{\mathcal{N}}\left[\frac{f(x+\rho v)-f(x-\rho v)}{2\rho}v\right],$$

which is precisely the vanilla zeroth-order gradient estimator with Gaussian perturbation.

### B.2 PROOF OF THEOREM 3.1 (BALL)

*Proof.* Recall that under the uniform ball perturbation, the $t$-SAM gradient is

$$\nabla_x F_t(x) = \frac{\mathbb{E}_{v\sim\mathcal{U}(\sqrt{d}\mathbb{B}^d)}[e^{tf(x+\rho v)}\nabla f(x+\rho v)]}{\mathbb{E}_{v\sim\mathcal{U}(\sqrt{d}\mathbb{B}^d)}[e^{tf(x+\rho v)}]}. \tag{18}$$

Denote $Z=\int_{\sqrt{d}\mathbb{B}^d}e^{tf(x+\rho v)}dv$. Then by definition, we have

$$\mathbb{E}_{v\sim\mathcal{U}(\sqrt{d}\mathbb{B}^d)}[e^{tf(x+\rho v)}] = \frac{\int_{\sqrt{d}\mathbb{B}^d}e^{tf(x+\rho v)}dv}{\mathrm{Vol}(\sqrt{d}\mathbb{B}^d)} = \frac{Z}{\mathrm{Vol}(\sqrt{d}\mathbb{B}^d)} \tag{19}$$

$$\mathbb{E}_{v\sim\mathcal{U}(\sqrt{d}\mathbb{B}^d)}[e^{tf(x+\rho v)}\nabla f(x+\rho v)] = \frac{\int_{\sqrt{d}\mathbb{B}^d}e^{tf(x+\rho v)}\nabla f(x+\rho v)dv}{\mathrm{Vol}(\sqrt{d}\mathbb{B}^d)}, \tag{20}$$

and applying them to Eq. (18) gives us

$$\nabla_x F_t(x) = \frac{\int_{\sqrt{d}\mathbb{B}^d} e^{tf(x+\rho v)} \nabla f(x+\rho v) dv}{Z}.$$

By change of variable, we have

$$\nabla_x \int_{\sqrt{d}\mathbb{B}^d} e^{tf(x+\rho v)} dv = \int_{\sqrt{d}\mathbb{B}^d} \nabla_x(e^{tf(x+\rho v)}) dv = \frac{1}{\rho} \int_{\sqrt{d}\mathbb{B}^d} \nabla_v(e^{tf(x+\rho v)}) dv.$$

According to the divergence theorem in higher dimensions, for a scalar field $\phi \in C^1 : \mathbb{R}^d \to \mathbb{R}$ and a compact volume $\Omega \subset \mathbb{R}^d$ with piecewise smooth boundary $\partial\Omega$, we have

$$\int_\Omega \nabla \phi \, dV = \int_{\partial\Omega} \phi \mathbf{n} \, dS \tag{21}$$

where $\mathbf{n}$ is the outward unit normal to the point on $\partial\Omega$, given that both sides of the equation are integrable over their domains. Therefore, by letting $\phi(v) = e^{tf(x+\rho v)}$, $\Omega = \sqrt{d}\mathbb{B}^d$, and $\partial\Omega = \sqrt{d}\mathbb{S}^{d-1}$, we obtain

$$\int_{\sqrt{d}\mathbb{B}^d} \nabla_v(e^{tf(x+\rho v)}) dv = \int_{\sqrt{d}\mathbb{S}^{d-1}} e^{tf(x+\rho u)} \frac{u}{\|u\|} du = \frac{1}{\sqrt{d}} \int_{\sqrt{d}\mathbb{S}^{d-1}} e^{tf(x+\rho u)} u \, du.$$

Expanding the LHS gives us

$$\int_{\sqrt{d}\mathbb{B}^d} e^{tf(x+\rho v)} \nabla_v f(x+\rho v) dv = \frac{1}{t\sqrt{d}} \int_{\sqrt{d}\mathbb{S}^{d-1}} e^{tf(x+\rho u)} u \, du. \tag{22}$$

Combining the above, we obtain

$$\nabla_x F_t(x) = \frac{1}{\rho Z} \int_{\sqrt{d}\mathbb{B}^d} e^{tf(x+\rho v)} \nabla_v f(x+\rho v) dv$$

$$\overset{(22)}{=} \frac{1}{t\rho\sqrt{d}Z} \int_{\sqrt{d}\mathbb{S}^{d-1}} e^{tf(x+\rho u)} u \, du$$

$$\overset{(a)}{=} \frac{\text{Area}(\sqrt{d}\mathbb{S}^{d-1})}{t\rho\sqrt{d}Z} \mathbb{E}_{u \sim \mathcal{U}(\sqrt{d}\mathbb{S}^{d-1})}[e^{tf(x+\rho u)} u]$$

$$\overset{(b)}{=} \frac{\sqrt{d} \cdot \text{Vol}(\sqrt{d}\mathbb{B}^d)}{t\rho\sqrt{d}Z} \mathbb{E}_{u \sim \mathcal{U}(\sqrt{d}\mathbb{S}^{d-1})}[e^{tf(x+\rho u)} u]$$

$$\overset{(19)}{=} \frac{1}{t\rho} \frac{\mathbb{E}_{u \sim \mathcal{U}(\sqrt{d}\mathbb{S}^{d-1})}[e^{tf(x+\rho u)} u]}{\mathbb{E}_{v \sim \mathcal{U}(\sqrt{d}\mathbb{B}^d)}[e^{tf(x+\rho v)}]} \tag{23}$$

where $(a)$ follows the definition of $\mathbb{E}_{u \sim \mathcal{U}(\sqrt{d}\mathbb{S}^{d-1})}[e^{tf(x+\rho u)} u]$ and $(b)$ is due to $\text{Area}(r\mathbb{S}^{d-1}) = \frac{d}{r} \cdot \text{Vol}(r\mathbb{B}^d)$, which gives us $\text{Area}(\sqrt{d}\mathbb{S}^{d-1}) = \sqrt{d} \cdot \text{Vol}(\sqrt{d}\mathbb{B}^d)$.

**Case of $t \to 0$.** As $t \to 0$, we apply L'Hôpital's rule to obtain

$$\lim_{t \to 0} \nabla_x F_t(x) = \frac{\lim_{t \to 0} \mathbb{E}_{\mathcal{S}}[e^{tf(x+\rho u)} f(x+\rho u) u]}{\lim_{t \to 0} \rho \mathbb{E}_{\mathcal{B}}[e^{tf(x+\rho v)}] + t\rho \mathbb{E}_{\mathcal{B}}[e^{tf(x+\rho v)} f(x+\rho v)]}$$

$$= \mathbb{E}_{\mathcal{S}}\left[\frac{f(x+\rho u) u}{\rho}\right]$$

$$= \mathbb{E}_{\mathcal{S}}\left[\frac{f(x+\rho u) u}{2\rho}\right] + \mathbb{E}_{\mathcal{S}}\left[\frac{f(x+\rho(-u))(-u)}{2\rho}\right]$$

$$= \mathbb{E}_{\mathcal{S}}\left[\frac{f(x+\rho u) - f(x-\rho u)}{2\rho} u\right],$$

which recovers the vanilla zeroth-order gradient estimator in Eq. (13).

$\square$

### B.3 SPHERE PERTURBATION REUSING

In Theorem 3.1, we use Eq. (5) to approximate (4). The rationale of this choice is the fact that most of the volume of a high-dimensional ball is concentrated near its boundary. As specified in Lemma B.1, $\mathbb{E}[\|v\|] \approx \sqrt{d}$ and $\mathrm{Var}(\|v\|) \approx \frac{1}{3d}$ for $d \gg 1$, which agrees with what we encounter in practice. This allows us to use the same sampled perturbations and the computed losses to compute both the numerator and the denominator, which thus gives ZEST the same computational workload as the vanilla zeroth-order optimization method.

**Lemma B.1** (Measure of Concentration). *For a random point uniformly sampled from a ball with radius $\sqrt{d}$, its norm $\|v\|$ satisfies*

$$\mathbb{E}[\|v\|] = \sqrt{d}\left(1 - \frac{1}{d+1}\right) \tag{24}$$

$$\mathrm{Var}(\|v\|) = \frac{d^2}{d+2} - \frac{d^3}{(d+1)^2} \stackrel{d \gg 1}{\approx} \frac{1}{3d} \tag{25}$$

*Proof.* Denote $q_{\|v\|}(r)$ as the probability density of the event $\|v\| = r$, which is proportional to the surface area of the sphere $r\mathbb{S}^{d-1}$:

$$q_{\|v\|}(r) = \frac{dr^{d-1}}{d^{d/2}}, 0 \le r \le \sqrt{d}.$$

Then the first and second moment of $\|v\|$ are

$$\mathbb{E}[\|v\|] = \int_0^{\sqrt{d}} r \cdot q_{\|v\|}(r) dr = \frac{d}{d^{d/2}} \int_0^{\sqrt{d}} r^d dr = \sqrt{d}\frac{d}{d+1}$$

$$\mathbb{E}[\|v\|^2] = \int_0^{\sqrt{d}} r^2 \cdot q_{\|v\|}(r) dr = \frac{d}{d^{d/2}} \int_0^{\sqrt{d}} r^{d+1} dr = \frac{d^2}{d+2}$$

and thus

$$\mathrm{Var}(\|v\|) = \frac{d^2}{d+2} - \frac{d^3}{(d+1)^2} = \frac{d^2}{(d+2)(d+1)^2} \stackrel{d \to \infty}{\longrightarrow} \lim_{d \to \infty} \frac{1}{3d+4}.$$

$\square$

Apart from the intuition that sampling from the ball is very similar to sampling from the sphere in high-dimensional space, we also explicitly quantify the bias between Eq. (4) and (5) from Theorem 3.1 in the following Theorem B.2.

**Theorem B.2** (Bias of Using $\mathbb{E}_{\mathcal{S}}$ in Denominator). *Assume that $f(x;\xi)$ is $L$-smooth and $M$-Lipschitz for any $x,\xi$. We have two estimators*

$$\nabla(x) = \frac{1}{t\rho} \frac{\overbrace{\mathbb{E}_{\mathcal{S}}[(e^{tf(x+\rho u)} - e^{tf(x-\rho u)})u]}^{N_{\mathcal{S}}}}{\underbrace{\mathbb{E}_{\mathcal{B}}[e^{tf(x+\rho v)} + e^{tf(x-\rho v)}]}_{D_{\mathcal{B}}}} \quad and \quad \tilde{\nabla}(x) = \frac{1}{t\rho} \frac{N_{\mathcal{S}}}{\underbrace{\mathbb{E}_{\mathcal{S}}[e^{tf(x+\rho u)} + e^{tf(x-\rho u)}]}_{D_{\mathcal{S}}}},$$

*and the bias from replacing $D_{\mathcal{B}}$ with $D_{\mathcal{S}}$ in the denominator is*

$$Bias(x) = \nabla(x) - \tilde{\nabla}(x) = \frac{N_{\mathcal{S}}}{t\rho}\left(\frac{1}{D_{\mathcal{B}}} - \frac{1}{D_{\mathcal{S}}}\right) = \frac{N_{\mathcal{S}}}{t\rho}\left(\frac{D_{\mathcal{S}} - D_{\mathcal{B}}}{D_{\mathcal{B}}D_{\mathcal{S}}}\right). \tag{26}$$

*With choosing $\rho \le d^{-\frac{4}{5}}$, we have*

$$\|Bias(x)\| \le O\left(\frac{1}{\sqrt{d}}\right).$$

*Proof.* Let $u$ be uniform on the sphere with radius $\sqrt{d}$ and write a point drawn uniformly from the ball as

$$v=ru, \quad r\in[0,1] \quad \text{with density} \quad p(r)=dr^{d-1}.$$

Since we have

$$\mathbb{E}[r^n]=\int_0^1 r^n p(r)dr=d\int_0^1 r^{r+d-1}dr=\frac{d}{d+n}=1-\frac{n}{d+n}$$

and $r\to 1$ in probability as $d\to\infty$. Then we have

$$D_{\mathcal{B}}=\mathbb{E}_r\mathbb{E}_u[e^{tf(x+\rho ru)}+e^{tf(x-\rho ru)}] \quad \text{and} \quad D_{\mathcal{S}}=\mathbb{E}_u[e^{tf(x+\rho u)}+e^{tf(x-\rho u)}].$$

Denote $g:=\nabla f(x)$ and $H:=\nabla^2 f(x)$. The Taylor expansion of $f(x\pm\rho ru)$ and $e^z$ gives

$$f(x\pm\rho ru)=f(x)\pm\rho rg^\top u+\frac{\rho^2 r^2}{2}u^\top Hu+\sum_{n\geq 3}O\left(\frac{\rho^n r^n}{n!}\|u\|^n\right), \tag{27}$$

$$e^z=1+z+\frac{z^2}{2}+\sum_{n\geq 3}\frac{z^n}{n!}. \tag{28}$$

So we have

$$\exp(tf(x\pm r\rho u))=\exp(tf(x))\exp\left(\pm t\rho rg^\top u+\frac{t\rho^2 r^2}{2}u^\top Hu+\sum_{n\geq 3}O\left(\frac{t\rho^n r^n}{n!}\|u\|^n\right)\right)$$

and further apply Eq. (27) and (28) to obtain nested Taylor expansion as

$$\exp\left(\pm t\rho rg^\top u+\frac{t\rho^2 r^2}{2}u^\top Hu+\sum_{n\geq 3}O\left(\frac{t\rho^n r^n}{n!}\|u\|^n\right)\right)$$

$$=1\pm t\rho rg^\top u+\frac{t\rho^2 r^2}{2}u^\top Hu+\sum_{n\geq 3}O\left(\frac{t\rho^n r^n}{n!}\|u\|^n\right)+\frac{t^2\rho^2 r^2(g^\top u)^2}{2}+...$$

$$=1\pm t\rho rg^\top u+\frac{t\rho^2 r^2}{2}u^\top Hu+\frac{t^2\rho^2 r^2(g^\top u)^2}{2}+\sum_{n\geq 3}O\left(\frac{t\rho^n r^n}{n^s}\|u\|^n\right)$$

where in the last step we absorb terms of order $n\geq 3$ into $O(\cdot)$ and leverage the combinatorial fact that for any $0<s\leq\log_3 4\approx 1.262$, $m!\prod_{i\in[m]}a_i!\geq n^s$ holds for any $n\geq 3$ where $n=\sum_{i\in[m]}a_i, a_i\in\mathbb{Z}_{\geq 0}, m\leq n$. Therefore, we take $s=\log_3 4$ and have

$$D_{\mathcal{B}}=2e^{tf(x)}\left[1+\frac{t\rho^2\mathbb{E}_r[r^2]}{2}\mathbb{E}_u[u^\top Hu]+\frac{t^2\rho^2\mathbb{E}_r[r^2]\mathbb{E}_u[(g^\top u)^2]}{2}+\sum_{n\geq 3}O\left(\frac{t\rho^n\mathbb{E}_r[r^n]}{n^s}\mathbb{E}_u[\|u\|^n]\right)\right]$$

$$=2e^{tf(x)}\left[1+\frac{t\rho^2\text{Tr}(H)}{2}\left(1-\frac{2}{d+2}\right)+\frac{t^2\rho^2\|g\|^2}{2}\left(1-\frac{2}{d+2}\right)+\sum_{n\geq 3}O\left(\frac{t\rho^n}{n^s}d^{\frac{n}{2}}\left(1-\frac{n}{d+n}\right)\right)\right]$$

and

$$D_{\mathcal{S}}=2e^{tf(x)}\left[1+\frac{t\rho^2\mathbb{E}_u[u^\top Hu]}{2}+\frac{t^2\rho^2\mathbb{E}_u[(g^\top u)^2]}{2}+\sum_{n\geq 3}O\left(\frac{t\rho^n}{n^s}\mathbb{E}_u[\|u\|^n]\right)\right]$$

$$=2e^{tf(x)}\left[1+\frac{t\rho^2\text{Tr}(H)}{2}+\frac{t^2\rho^2\|g\|^2}{2}+\sum_{n\geq 3}O\left(\frac{t\rho^n}{n^s}d^{\frac{n}{2}}\right)\right].$$

Since each term of $D_{\mathcal{B}}$ corresponds to one term in $D_{\mathcal{S}}$ that differ by a scale $(1-\frac{n}{d+n})$ for their order $n$, we have

$$D_{\mathcal{S}}-D_{\mathcal{B}}=2e^{tf(x)}\left[\frac{t\rho^2\text{Tr}(H)}{d+2}+\frac{t^2\rho^2\|g\|^2}{d+2}+\sum_{n\geq 3}O\left(\frac{t\rho^n n}{n^s(d+n)}d^{\frac{n}{2}}\right)\right]$$

and

$$|D_\mathcal{S}-D_\mathcal{B}|\leq 2e^{tf(x)}\left[t\rho^2 L+\frac{t^2\rho^2\|g\|^2}{d}+\sum_{n\geq 3}O\left(\frac{t\rho^n}{n^s}d^{\frac{n}{2}}\right)\right]$$

due to $f$ being $L$-smooth and thus $\mathrm{Tr}(H)\leq\|H\|_{\mathrm{op}}d=Ld$. Additionally,

$$e^{tf(x+\rho u)}-e^{tf(x-\rho u)}=2e^{tf(x)}\left[t\rho r g^\top u+\sum_{n\geq 3}O\left(\frac{t\rho^n r^n}{n^s}\|u\|^n\right)\right]$$

and thus we have

$$\|N_\mathcal{S}\|\leq 2e^{tf(x)}\left[t\rho\mathbb{E}_r[r]\big\|\mathbb{E}_u[(g^\top u)u]\big\|+\sum_{n\geq 3}O\left(\frac{t\rho^n\mathbb{E}_r[r^n]}{n^s}d^{\frac{n+1}{2}}\right)\right]$$

$$=2e^{tf(x)}\left[t\rho\frac{d}{d+1}\|g\|+\sum_{n\geq 3}O\left(\frac{t\rho^n d}{n^s(d+n)}d^{\frac{n+1}{2}}\right)\right]$$

$$\leq 2e^{tf(x)}\left[t\rho\|g\|+\sum_{n\geq 3}O\left(\frac{t\rho^n}{n^s}d^{\frac{n+1}{2}}\right)\right].$$

If we choose $\rho\leq d^{-\frac{4}{5}}$, we have $\rho^n d^{\frac{n+1}{2}}\leq d^{\frac{1}{2}-\frac{3n}{10}}\leq d^{-\frac{2}{5}}$ and $\rho^{n-1}d^{\frac{n}{2}}\leq d^{\frac{4}{5}-\frac{3n}{10}}\leq d^{-\frac{1}{10}}$ for $n\geq 3$. Additionally, our loss $f\geq 0$ and thus $D_\mathcal{S},D_\mathcal{B}\geq 2$. Applying them to Eq. (26), we have

$$\|\mathrm{Bias}(x)\|=\frac{\|N_\mathcal{S}\|}{t\rho}\left(\frac{|D_\mathcal{S}-D_\mathcal{B}|}{D_\mathcal{B}D_\mathcal{S}}\right)$$

$$\leq\frac{\|N_\mathcal{S}\|}{4t\rho}|D_\mathcal{S}-D_\mathcal{B}|$$

$$\leq\frac{(e^{tf(x)})^2}{t\rho}\left[t\rho\|g\|+\sum_{n\geq 3}O\left(\frac{t\rho^n}{n^s}d^{\frac{n+1}{2}}\right)\right]\left[t\rho^2 L+\frac{t^2\rho^2\|g\|^2}{d}+\sum_{n\geq 3}O\left(\frac{t\rho^n}{n^s}d^{\frac{n}{2}}\right)\right]$$

$$=(e^{tf(x)})^2\left[t\rho\|g\|+\sum_{n\geq 3}O\left(\frac{t\rho^n}{n^s}d^{\frac{n+1}{2}}\right)\right]\left[\rho L+\frac{t\rho\|g\|^2}{d}+\sum_{n\geq 3}O\left(\frac{\rho^{n-1}}{n^s}d^{\frac{n}{2}}\right)\right]$$

$$\overset{(a)}{\leq}(e^{tf(x)})^2\left[\frac{t\|g\|}{d^{\frac{4}{5}}}+d^{-\frac{2}{5}}t\sum_{n\geq 3}O\left(\frac{1}{n^s}\right)\right]\left[\frac{L}{d^{\frac{4}{5}}}+\frac{t\|g\|^2}{d^{\frac{9}{5}}}+d^{-\frac{1}{10}}\sum_{n\geq 3}O\left(\frac{1}{n^s}\right)\right]$$

$$\overset{(b)}{\leq}(e^{tf(x)})^2\left[\frac{t\|g\|}{d^{\frac{4}{5}}}+tCd^{-\frac{2}{5}}\right]\left[\frac{L}{d^{\frac{4}{5}}}+\frac{t\|g\|^2}{d^{\frac{9}{5}}}+Cd^{-\frac{1}{10}}\right]$$

$$=O\left(\frac{1}{\sqrt{d}}\right)$$

where $(a)$ is due to setting $\rho\leq d^{-\frac{4}{5}}$ and $(b)$ follows that $\sum_n\frac{1}{n^s}$ converges to some constant $C$ if $s>1$. $\qquad\square$

## B.4 BIAS-CORRECTED RATIO ESTIMATE

In this section, we derive the bias-corrected estimate with bias $O(1/k^2)$. Recall that in each iteration, we sample $k$ perturbations and compute $a_i^+=e^{tf(x+\rho v_i)}$, $a_i^-=e^{tf(x-\rho v_i)}$, and $Z=\sum_{i=1}^k a_i^++a_i^-$. We aim to approximate

$$\frac{\mathbb{E}[A]}{\mathbb{E}[B]}, \text{ with samples } A_i=(a_i^+-a_i^-)v_i \text{ and } B_i=a_i^++a_i^-, i\in[k].$$

In the following, we show that making up the bias in the naive plug-in leads to the following estimate:

$$t\rho G_{\text{BC}}^k = \sum_{i=1}^k \left\{ 1 + \frac{k}{k-1}[\bar{a}_i^+ + \bar{a}_i^- - \sum_{i=1}^k (\bar{a}_i^+ + \bar{a}_i^-)^2] \right\} (\bar{a}_i^+ - \bar{a}_i^-)v_i.$$

*Proof.* Define the function $g(x,y) = \frac{x}{y}$ with $x \in \mathbb{R}^d$ and $y \in \mathbb{R}$. Let $\bar{A} = \frac{1}{k}\sum_{i=1}^k A_i$, $\bar{B} = \frac{1}{k}\sum_{i=1}^k B_i$, $\mu_A = \mathbb{E}[A]$, and $\mu_B = \mathbb{E}[B]$. We expand $g(\bar{A}, \bar{B})$ around the point $(\mu_A, \mu_B)$ and have

$$g(\bar{A}, \bar{B}) \approx g(\mu_A, \mu_B) + g_{\bar{A}}^\top (\bar{A} - \mu_A) + g_{\bar{B}}(\bar{B} - \mu_B)$$
$$+ \frac{1}{2}[(\bar{A} - \mu_A)^\top g_{\mu_A \mu_A}(\bar{A} - \mu_A) + 2(\bar{A} - \mu_A)^\top g_{\mu_A \mu_B}(\bar{B} - \mu_B) + g_{\mu_B \mu_B}(\bar{B} - \mu_B)^2]$$

where $g_{\bar{A}} = \frac{\partial g(\bar{A}, \bar{B})}{\partial \bar{A}}$, $g_{\bar{B}} = \frac{\partial g(\bar{A}, \bar{B})}{\partial \bar{B}}$, $g_{\mu_A \mu_A} = \frac{\partial^2 g(\mu_A, \mu_B)}{\partial \mu_A^2} = 0$, $g_{\mu_B \mu_B} = \frac{\partial^2 g(\mu_A, \mu_B)}{\partial \mu_B^2} = \frac{2\mu_A}{\mu_B^3}$, and $g_{\mu_A \mu_B} = \frac{\partial^2 g(\mu_A, \mu_B)}{\partial \mu_A \partial \mu_B} = -\frac{1}{\mu_B^2}$. Applying them to the approximate equality and taking the expectation on both sides, we have

$$\mathbb{E}\left[\frac{\bar{A}}{\bar{B}}\right] \approx \frac{\mu_A}{\mu_B} - \frac{1}{\mu_B^2}\mathbb{E}[(\bar{A} - \mu_A)^\top(\bar{B} - \mu_B)] + \frac{\mu_A}{\mu_B^3}\mathbb{E}[(\bar{B} - \mu_B)^2]$$
$$= \frac{\mu_A}{\mu_B} - \frac{1}{\mu_B^2}\text{Cov}(\bar{A}, \bar{B}) + \frac{\mu_A}{\mu_B^3}\text{Var}(\bar{B})$$
$$= \frac{\mathbb{E}[A]}{\mathbb{E}[B]} - \frac{1}{\mu_B^2 k}\text{Cov}(A, B) + \frac{\mu_A}{\mu_B^3 k}\text{Var}(B).$$

Therefore, we use

$$t\rho G_{\text{BC}}^k = \frac{\bar{A}}{\bar{B}} + \frac{1}{k(k-1)}\left[ \frac{\sum_{i=1}^k (A_i - \bar{A})(B_i - \bar{B})}{\bar{B}^2} - \frac{\bar{A}\sum_{i=1}^k (B_i - \bar{B})^2}{\bar{B}^3} \right].$$

Denote $a_i^+ = e^{tf(x+\rho v_i)}$, $a_i^- = e^{tf(x-\rho v_i)}$ and thus $A_i = (a_i^+ - a_i^-)v_i$ and $B_i = a_i^+ + a_i^-$. In practice, we record $Z = \sum_{i=1}^k a_i^+ + a_i^- = k\bar{B}$ and work with the normalized values $\bar{a}_i^+ := a_i^+/Z$ and $\bar{a}_i^- := a_i^-/Z$ for numerical stability. So we re-express $t\rho G_{\text{BC}}^k$ with $A_i' := (\bar{a}_i^+ - \bar{a}_i^-)v_i$, $B_i' := \bar{a}_i^+ + \bar{a}_i^-$, and thus $\bar{A}' := \frac{1}{k}\sum_{i=1}^k A_i'$ as

$$t\rho G_{\text{BC}}^k = \sum_{i=1}^k A_i' + \sum_{i=1}^k \frac{(kB_i' - 1)}{k-1}(A_i' - \bar{A}') - \bar{A}'\sum_{i=1}^k \frac{(kB_i' - 1)^2}{k-1}$$
$$= \sum_{i=1}^k \left\{ 1 + \frac{k}{k-1}[B_i' - \sum_{i=1}^k (B_i')^2] \right\} A_i'.$$

$\square$

In Figure 3, we show that the bias-corrected estimator indeed has smaller bias. Recall that we aim to estimate $\frac{\mathbb{E}[A]}{\mathbb{E}[B]}$, where $A \in \mathbb{R}^d$ and $B \in \mathbb{R}$. We set $d = 100, A \sim \mathcal{N}(\mu_A, 64I_{100})$ with $\mu_A = [40, -40, 40, -40, ...] \in \mathbb{R}^{100}$ and $B \sim \mathcal{N}(80, 16)$. Denote the direct plug-in estimator as $G_N$ and the bias-corrected one as $G_{\text{BC}}$. Our simulation results show their bias and variance across different $k$ values. We observe that (1) both estimators correctly converge to the ground truth (i.e., error norm reduces to 0) as $k$ increases, (2) $G_{\text{BC}}$ has smaller bias when $k$ is small, and (3) they have approximately the same variance, which agrees with the theory (Van Kempen & Van Vliet, 2000b).

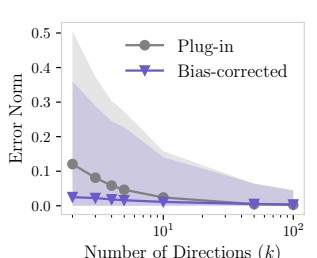

Figure 3: Error norms of the averaged estimate under 1000 trials and the standard deviation (shades) among the trials.

## B.5 DERIVATION OF SECTION 4.1

**Derivation of Eq. (10).** Recall that the Hessian $\nabla^2 f(x)$ is written as $\nabla^2 f(x) = Q^\top \Lambda Q$, where the orthogonal $Q$ has columns $\{e_1,...,e_d\}$ that are ordered eigenvectors of $\nabla^2 f(x)$, and $\Lambda = \text{diag}(\lambda_1, \lambda_2, ..., \lambda_d)$ where $\lambda_1 \geq ... \geq \lambda_d$ are the order eigenvalues of $\nabla^2 f(x)$. Observe that $u := Qv$ has the same distribution as $v$ since Gaussian is rotation-invariant. Denote $g := Q\nabla f(x)$ where $g_i$ is the component of the gradient along the $i$-th eigenvector. Then we have

$$
\mathbb{E}_{v \sim \mathcal{N}}\left[\exp\left(t\left(\rho \nabla f(x)^\top v + \frac{\rho^2}{2} v^\top \nabla^2 f(x) v\right)\right)\right]
$$

$$
= \mathbb{E}_{u \sim \mathcal{N}}\left[\exp\left(t\left(\rho g^\top u + \frac{\rho^2}{2} u^\top \Lambda u\right)\right)\right]
$$

$$
= (2\pi)^{-d/2} \int \exp\left(t(\rho g^\top u + \frac{\rho^2}{2} u^\top \Lambda u) - \frac{1}{2}\sum_{i=1}^{d} u_i^2\right) du
$$

$$
= (2\pi)^{-d/2} \int \exp\left(t\rho \sum_{i=1}^{d} g_i u_i + \sum_{i=1}^{d} \frac{t\rho^2 \lambda_i - 1}{2} u_i^2\right) du
$$

$$
= \prod_{i=1}^{d} \int \frac{1}{\sqrt{2\pi}} \exp\left(-\frac{1 - t\rho^2 \lambda_i}{2} u_i^2 + t\rho g_i u_i\right) du_i
$$

$$
= \prod_{i=1}^{d} \int \frac{1}{\sqrt{2\pi}} \exp\left(\left[-\frac{1 - t\rho^2 \lambda_i}{2}\left(u_i - \frac{t\rho g_i}{1 - t\rho^2 \lambda_i}\right)^2 + \frac{(t\rho g_i)^2}{2(1 - t\rho^2 \lambda_i)}\right]\right) du_i
$$

$$
= \prod_{i=1}^{d} \exp\left(\frac{(t\rho g_i)^2}{2(1 - t\rho^2 \lambda_i)}\right) \int \frac{1}{\sqrt{2\pi}} \exp\left(\left[-\frac{1 - t\rho^2 \lambda_i}{2}\left(u_i - \frac{t\rho g_i}{1 - t\rho^2 \lambda_i}\right)^2\right]\right) du_i
$$

$$
= \prod_{i=1}^{d} \frac{\exp\left(\frac{(t\rho g_i)^2}{2(1 - t\rho^2 \lambda_i)}\right)}{\sqrt{1 - t\rho^2 \lambda_i}} \underbrace{\int \frac{\sqrt{1 - t\rho^2 \lambda_i}}{\sqrt{2\pi}} \exp\left(\left[-\frac{1 - t\rho^2 \lambda_i}{2}\left(u_i - \frac{t\rho g_i}{1 - t\rho^2 \lambda_i}\right)^2\right]\right) du_i}_{=1}
$$

$$
= \frac{\exp\left(\sum_{i=1}^{d} \frac{(t\rho g_i)^2}{2(1 - t\rho^2 \lambda_i)}\right)}{\prod_{i=1}^{d} \sqrt{1 - t\rho^2 \lambda_i}}
$$

where $du$ denotes the Lebesgue measure on $\mathbb{R}^d$. Note that it is required for any $i$, $1 - t\rho^2 \lambda_i > 0$, i.e., choose $t\rho^2 < \frac{1}{\lambda_{\max}}$ if $\lambda_{\max} > 0$. The regularizer is thus

$$
R_t = \frac{1}{t} \sum_{i=1}^{d} \frac{(t\rho g_i)^2}{2(1 - t\rho^2 \lambda_i)} - \frac{1}{2t} \sum_{i=1}^{d} \log(1 - t\rho^2 \lambda_i)
$$

$$
= \frac{1}{2t} \sum_{i=1}^{d} \left[\frac{(t\rho g_i)^2}{1 - t\rho^2 \lambda_i} - \log(1 - t\rho^2 \lambda_i)\right].
$$

When $t \to 0$, we apply L'Hôpital's rule to obtain

$$
\lim_{t \to 0} R_t = -\sum_{i=1}^{d} \lim_{t \to 0} \frac{d(\log(1 - t\rho^2 \lambda_i))/dt}{d(2t)/dt} = -\sum_{i=1}^{d} \lim_{t \to 0} \frac{-\rho^2 \lambda_i}{2(1 - t\rho^2 \lambda_i)} = \frac{\rho^2}{2} \sum_{i=1}^{d} \lambda_i.
$$

## B.6 DERIVATION OF SECTION 4.2

**Derivation of Eq. (11)** Recall that we denote $g = Q\nabla^2 f(x)$ and $\Lambda = \text{diag}(\lambda_1, \lambda_2, ..., \lambda_d)$ defined as before. Assume that $\|\nabla f(x)\| < \infty$ and $\nabla^2 f(x)$ has bounded eigenvalues for any $x$ in our optimization trajectory.

We denote $X = \rho g^\top u + \frac{\rho^2}{2} u^\top \Lambda u$ with $X < \infty$, and we thus have $\mathbb{E}[\exp(tX)] < \infty$ for any $t < \infty$. Since $h(X) = \frac{1}{t} \log(\mathbb{E}[\exp(tX)])$ is continuous for $t \in \{t : \mathbb{E}[\exp(tX)] < \infty\}$, $h(X)$ is continuous and non-decreasing in $t$ for any $t < \infty$. Furthermore, the regularizer sensitivity is

$$\phi_i = \frac{1}{t} \cdot \frac{1}{\mathbb{E}[\exp(tX)]} \cdot \frac{\partial \mathbb{E}[\exp(tX)]}{\partial \lambda_i} = \frac{\rho^2}{2} \frac{\mathbb{E}[\exp(t[\rho g^\top u + \frac{\rho^2}{2} u^\top \Lambda u]) u_i^2]}{\mathbb{E}[\exp(t[\rho g^\top u + \frac{\rho^2}{2} u^\top \Lambda u])]}.$$

It is continuous and non-decreasing in $\lambda_i$ due to

$$\frac{\partial \phi_i}{\partial \lambda_i} = \frac{t\rho^4}{4} \cdot \frac{\mathbb{E}[u_i^4 e^{tX}] \mathbb{E}[e^{tX}] - (\mathbb{E}[u_i^2 e^{tX}])^2}{(\mathbb{E}[e^{tX}])^2} \geq 0$$

where the inequality follows the Cauchy-Schwarz inequality $(\mathbb{E}[AB])^2 \leq \mathbb{E}[A^2]\mathbb{E}[B^2]$. Therefore, it suffices to analyze $R_t$ and $\phi_i(t)$ under the two extreme cases that $t \to 0$ and $\infty$. Recall that we have

$$R_t = \frac{1}{t} \log\left( \int \exp\left( t\rho g^\top u + \frac{t\rho^2}{2} u^\top \Lambda u \right) d\mu(u) \right) \quad \text{with } u = Qv \sim \mathcal{U}(\sqrt{d}\mathbb{B}^d)$$

$$= \frac{1}{t} \log\left( \frac{1}{\text{Vol}(\sqrt{d}\mathbb{B}^d)} \int_{\|u\| \leq \sqrt{d}} \exp\left( t\rho g^\top u + \frac{t\rho^2}{2} u^\top \Lambda u \right) du \right). \tag{29}$$

When $t \to 0$, we apply L'Hôpital's rule to Eq. (29) and obtain

$$\lim_{t \to 0} R_t = \lim_{t \to 0} \frac{\int_{\|u\| \leq \sqrt{d}} \nabla_t \left[ \exp\left( t\rho g^\top u + \frac{t\rho^2}{2} u^\top \Lambda u \right) \right] du}{\int_{\|u\| \leq \sqrt{d}} \exp\left( t\rho g^\top u + \frac{t\rho^2}{2} u^\top \Lambda u \right) du}$$

$$= \lim_{t \to 0} \frac{\int_{\|u\| \leq \sqrt{d}} \exp\left( t\rho g^\top u + \frac{t\rho^2}{2} u^\top \Lambda u \right) \left( \rho g^\top u + \frac{\rho^2}{2} u^\top \Lambda u \right) du}{\int_{\|u\| \leq \sqrt{d}} \exp\left( t\rho g^\top u + \frac{t\rho^2}{2} u^\top \Lambda u \right) du}$$

$$= \lim_{t \to 0} \frac{\int_{\|u\| \leq \sqrt{d}} \rho g^\top u + \frac{\rho^2}{2} u^\top \Lambda u\, du}{\text{Vol}(\sqrt{d}\mathbb{B}^d)}$$

$$= \frac{\rho^2}{2\text{Vol}(\sqrt{d}\mathbb{B}^d)} \lim_{t \to 0} \int_{\|u\| \leq \sqrt{d}} u^\top \Lambda u\, du$$

$$= \frac{\rho^2 d}{2(d+2)} \sum_{i=1}^{d} \lambda_i$$

where the last step is due to

$$\int_{\|u\| \leq \sqrt{d}} u^\top \Lambda u\, du = \int_{\|u\| \leq \sqrt{d}} \left( \sum_{i=1}^{d} \lambda_i u_i^2 \right) du = \sum_{i=1}^{d} \lambda_i \int_{\|u\| \leq \sqrt{d}} u_i^2\, du$$

and

$$\int_{\|u\| \leq \sqrt{d}} u_i^2\, du = \frac{1}{d} \int_{\|u\| \leq \sqrt{d}} \|u\|^2\, du = \frac{d}{2(d+2)} \text{Vol}(\sqrt{d}\mathbb{B}^d).$$

**Derivation of Eq. (12).** When $t \to \infty$, we apply Laplace's principle (Dembo (2009)) that for a Lebesgue-measurable set $\mathcal{M} \in \mathbb{R}^d$ and a measurable function $\varphi : \mathbb{R}^d \to \mathbb{R}$ that satisfy $\int_{\mathcal{M}} e^{\varphi(x)} dx < \infty$, we have

$$\lim_{t \to \infty} \frac{1}{t} \log \int_{\mathcal{M}} e^{t\varphi(x)} dx = \max_{x \in \mathcal{M}} \varphi(x).$$

Let $\varphi(u) = \rho g^\top u + \frac{\rho^2}{2} u^\top \Lambda u$ and $\mathcal{M} = \sqrt{d}\mathbb{B}^d$. Since $\mathcal{M}$ is measurable and $\varphi(x) \leq \rho\sqrt{d}\|a\| + \frac{\rho^2 d}{2} \max(\lambda_{\max}, 0)$, the integrability condition satisfies, and we have

$$\lim_{t \to \infty} R_t = \lim_{t \to \infty} \frac{1}{t} \log\left( \frac{1}{\text{Vol}(\sqrt{d}\mathbb{B}^d)} \right) + \lim_{t \to \infty} \frac{1}{t} \log\left( \int_{\mathcal{M}} e^{t\varphi(u)} du \right)$$

$$= \max_{\|u\| \leq \sqrt{d}} \varphi(u).$$

### B.7 GENERAL REGIME ($t \to \infty$)

Recall that we work in the Hessian eigenbasis with $\Lambda = \text{diag}(\lambda_1, ..., \lambda_d)$ and $g = Q\nabla f(x)$. In the general regime where both the slope and curve penalties are active, we use KKT conditions to solve the maximization problem with an inequality constraint

$$\max_{u: \|u\| \leq \sqrt{d}} \varphi(u) = \rho g^\top u + \frac{\rho^2}{2} u^\top \Lambda u.$$

From the Lagrangian

$$\mathcal{L}(u, \omega) = \rho a^\top u + \frac{\rho^2}{2} u^\top \Lambda u - \omega(u^\top u - d), \quad \omega \geq 0,$$

we have

$$\rho g + \rho^2 \Lambda u - 2\omega u = 0 \Longleftrightarrow (\rho^2 \Lambda - 2\omega I)u = -\rho g$$

$$\omega(\|u\| - \sqrt{d}) = 0$$

$$\|u\| \leq \sqrt{d}$$

$$2\omega I - \rho^2 \Lambda \succeq 0$$

by stationarity, complementary slackness, primal feasibility, and dual feasibility, respectively.

**Interior case.** When $\nabla^2 f(x) \preceq 0$ and the unconstrained maximizer is feasible, we take $\omega = 0$ and thus $u^\star = -(1/\rho)\Lambda^{-1}g$ and

$$R_\infty = \varphi(u^\star) = -\frac{1}{2} g^\top \Lambda^{-1} g = -\frac{1}{2} \sum_{i=1}^d \frac{g_i^2}{\lambda_i} \quad (\lambda_i \leq 0),$$

which indicates that making $\lambda_i$ more negative reduces the penalty. We also have the regularizer sensitivity that increases as $\lambda_i$ increases:

$$\phi_i = \frac{\partial R_\infty}{\partial \lambda_i} = -\frac{1}{2} g_i^2 \frac{\partial}{\partial \lambda_i}\left(\frac{1}{\lambda_i}\right) = \frac{1}{2} \frac{g_i^2}{\lambda_i^2}.$$

**Boundary case.** In the boundary case ($\omega > 0$), the maximizer $u^\star$ solves the KKT system for

$$\max_{u: \|u\| = \sqrt{d}} \rho g^\top u + \frac{\rho^2}{2} u^\top \Lambda u.$$

The stationarity states $u = (2\omega I - \rho^2 \Lambda)^{-1}\rho g$, which is well-defined when $2\omega I - \rho^2 \Lambda \succ 0$, i.e., $\omega > \frac{\rho^2}{2}\lambda_{\max}$. The correct $\omega$ is chosen so that $\|u(\omega)\| = \sqrt{d}$ holds. Note that $\|u(\omega)\|$ is strictly decreasing in $\omega \in (\frac{\rho^2}{2}\lambda_{\max}, \infty)$ since

$$\|u(\omega)\|^2 = \sum_{i=1}^d \frac{\rho^2 g_i^2}{(2\omega - \rho^2 \lambda_i)^2}$$

is strictly decreasing from $\infty$ (assume that $g_1 \neq 0$) to 0. Therefore, there is a unique solution $\omega^\star > \max(\frac{\rho^2}{2}\lambda_{\max}, 0)$. Then we can compute $u^\star = (2\omega^\star I - \rho^2 \Lambda)^{-1}\rho g$ since

$$\rho g + \rho^2 \Lambda u^\star - 2\omega^\star u^\star = 0. \tag{S}$$

Next, we compute the regularizer sensitivity $\psi_i$. Since $\omega^\star$ and $u^\star$ are functions of $\lambda_i$, we differentiate both sides of (S) with respect to $\lambda_i$ and obtain

$$\rho^2 E_i u^\star + \rho^2 \Lambda \frac{du^\star}{d\lambda_i} - 2\frac{d\omega^\star}{d\lambda_i} u^\star - 2\omega^\star \frac{du^\star}{d\lambda_i} = 0,$$

where $E_i$ is a diagonal matrix with a 1 at entry $i$ and 0's elsewhere. Differentiating both sides of the constraint that $(u^\star)^\top u^\star = d$ gives

$$2(u^\star)^\top \frac{du^\star}{d\lambda_i} = 0. \tag{C}$$

Therefore, differentiating $\varphi(u^\star)$ leads to

$$\frac{d}{d\lambda_i}\varphi(u^\star)=\rho g^\top\frac{du^\star}{d\lambda_i}+\frac{\rho^2}{2}\Big((u^\star)^\top E_i u^\star+2(u^\star)^\top\Lambda\frac{du^\star}{d\lambda_i}\Big).$$

Using stationarity (S) to replace $\rho g$ by $2\omega^\star u^\star-\rho^2\Lambda u^\star$ gives

$$\rho g^\top\frac{du^\star}{d\lambda_i}=(2\omega^\star u^\star-\rho^2\Lambda u^\star)^\top\frac{du^\star}{d\lambda_i}=2\omega^\star(u^\star)^\top\frac{du^\star}{d\lambda_i}-\rho^2(u^\star)^\top\Lambda\frac{du^\star}{d\lambda_i}.$$

By (C), $(u^\star)^\top du^\star/d\lambda_i=0$, so the first term vanishes. Therefore,

$$\frac{d}{d\lambda_i}\varphi(u^\star)=-\rho^2(u^\star)^\top\Lambda\frac{du^\star}{d\lambda_i}+\frac{\rho^2}{2}(u^\star)^\top E_i u^\star+\rho^2(u^\star)^\top\Lambda\frac{du^\star}{d\lambda_i}$$

$$=\frac{\rho^2}{2}(u_i^\star)^2$$

$$=\frac{\rho^4 g_i^2}{2(2\omega^\star-\rho^2\lambda_i)^2}. \tag{30}$$

Therefore, the regularizer sensitivity $\phi_i(\omega^\star,\lambda_i)$ for an arbitrary $\lambda_i$ is

$$\phi_i(\omega^\star,\lambda_i)=\frac{d}{d\lambda_i}\varphi(u^\star)=\frac{\rho^4 g_i^2}{2D_i^2}. \tag{31}$$

where $D_j:=2\omega^\star-\rho^2\lambda_j>0$. Differentiating the sensitivity w.r.t. $\lambda_i$ informs us whether the sensitivity is constant across all eigenvalues as in the average-case SAM regularizer, or increases as in the worst-case SAM and $t$-SAM regularizer $R_\infty$. To proceed, we track how $\omega^\star$ shifts when $\lambda_i$ changes by implicitly differentiating the secular equation

$$\psi(\omega^\star,\{\lambda_j\}):=\sum_{j=1}^d\frac{\rho^2 g_j^2}{(2\omega^\star-\rho^2\lambda_j)^2}=d. \tag{32}$$

Treat $\psi$ as a function of two variables, including $\omega^\star(\lambda_i)$, the function value with parameter $\lambda_i$, and the parameter $\lambda_i$ itself. Differentiating both sides of Eq. (32) w.r.t. $\lambda_i$ leads to

$$\frac{\partial\psi}{\partial\omega^\star}\frac{\partial\omega^\star}{\partial\lambda_i}+\frac{\partial\psi}{\partial\lambda_i}=0\quad\Longrightarrow\quad\frac{\partial\omega^\star}{\partial\lambda_i}=-\frac{\partial\psi/\partial\lambda_i}{\partial\psi/\partial\omega^\star} \tag{33}$$

by the chain rule. Further, we differentiate $\psi$ w.r.t. $\omega^\star$ and $\lambda_i$ respectively and have

$$\frac{\partial\psi}{\partial\omega^\star}=-4\sum_{j=1}^d\frac{\rho^2 g_j^2}{D_j^3},\qquad\frac{\partial\psi}{\partial\lambda_i}=\frac{2\rho^4 g_i^2}{D_i^3}.$$

Therefore, we apply the above to Eq. (33) and obtain

$$\frac{d\omega^\star}{d\lambda_i}=\frac{\frac{\rho^2 g_i^2}{D_i^3}}{2\sum_{j=1}^d\frac{g_j^2}{D_j^3}}. \tag{34}$$

Now we can compute

$$\frac{\partial\phi}{\partial\lambda_i}=-\frac{\rho^4 g_i^2}{D_i^3}\frac{\partial D_i}{\partial\lambda_i}=-\frac{\rho^4 g_i^2}{D_i^3}\Big[2\frac{\partial\omega^\star}{\partial\lambda_i}-\rho^2\Big]=\frac{\rho^6 g_i^2}{D_i^3}\left[1-\frac{\frac{g_i^2}{D_i^3}}{\sum_{j=1}^d\frac{g_j^2}{D_j^3}}\right]\geq 0, \tag{35}$$

which indicates that the sensitivity of $R_\infty$ to any arbitrary $\lambda_i$ grows when $\lambda_i$ increases. This is in contrast with the average-loss SAM, where the influence of all the eigenvalues is always equal.

### B.8 Choosing $t$

Denote the random variable $X = \rho g^\top u + \frac{\rho^2}{2} u^\top \Lambda u, u \sim \mathcal{U}(\sqrt{d}\mathbb{B}^d)$ and note that $m \leq X \leq M$ with $m = \frac{\rho^2 d}{2}\min(\lambda_{\min},0) - \rho\sqrt{d}\|g\|$ and $M = \frac{\rho^2 d}{2}\max(\lambda_{\max},0) + \rho\sqrt{d}\|g\|$. By Hoeffding's lemma, for any $t \in \mathbb{R}$, $\mathbb{E}[e^{tX}] \leq \exp\left(t\mathbb{E}[X] + \frac{t^2(M-m)^2}{8}\right)$. Then by Jensen's inequality,

$$R_t = \frac{1}{t}\log(\mathbb{E}[e^{tX}]) \leq \mathbb{E}[X] + \frac{t(M-m)^2}{8}.$$

Therefore, to keep $R_t$ within $\varepsilon$ from the expectation $\mathbb{E}[X] = \frac{\rho^2 d}{2(d+2)}\sum_{i=1}^d \lambda_i$, which is the sharpness regularizer in the average-case SAM objective, it suffices to take

$$t \leq \frac{8\varepsilon}{(M-m)^2} \leq \frac{32\varepsilon}{\rho^2 d[\rho\sqrt{d}\max(|\lambda_{\max}|,|\lambda_{\min}|) + 4\|g\|]^2}. \tag{36}$$

Since $\rho$ is usually chosen as $\rho \leq \sqrt{d}$ in zeroth-order optimization, the effective range of $t$ is $d$-independent. The remaining parameters, such as $\lambda_{\max}$, are problem-dependent, similar to the generalization bounds presented in prior literature (Li et al., 2024; Aminian et al., 2025). Therefore, in practice, one needs to find the $t$ that yields the best validation performance on the task of interest. It is not surprising that the vanilla SAM (i.e., $t \to \infty$) does not necessarily yield the optimal sharpness notion: Note that SAM is a min-max objective that can be too pessimistic and difficult to solve due to its non-smoothness. It considers the worst-case loss in the neighborhood, whereas there can be many bad points in the neighborhood that incur large losses that need to be considered.

Through our experiments with RoBERTa-Base under $t = \{0,1,5,20\}$, however, we observe that $t=1$ is a safe go-to choice for a preliminary trial since it almost always yields superior performance to $t=0$ (Figure 4). We find that $t=1$ matches or outperforms MeZO on 7 out of 8 settings by 1.7%, 0.8%, 0.4%, 2.9%, 5.7%, and 0.8%; the only setting when $t=1$ underperforms is on SST-5 by 0.1%.

### B.9 Low-Dimensional Examples

**Linear regime.** We generate the piecewise-linear $f$ by discretizing the function value surface of $h(x,y) = 0.07(8x^2 + 10y^2) + 0.14$. By forming $q$ triangles (i.e., planes) $\{P_j\}_{j \in [q]}$ that intersect with $h(x,y)$, we obtain $f(x,y) := \min_j P_j(x,y)$ that is piecewise-linear as desired. For the experiments, we run both zeroth-order methods with $k=500$ and $\rho=0.5$ for 40 iterations. We use $t=1$.

**Stationary regime.** We define $f: \mathbb{R}^2 \to \mathbb{R}$ by $f(x,y) = \frac{1}{5}[(x^2-1)^2 + \frac{1}{2}x(x^2-1)^2 + (1+2(1-x))y^2]$. In the experiments, we consistently start from the initialization at $(0,1)$. We run gradient descent for 50 iterations and zeroth-order methods with $k=500$ and $\rho=0.8$ for 100 iterations. We use $t=1$.

## C Experiments

In this section, we present our experiment setup and the additional results, including sharpness measurements and the results under different $t$.

### C.1 Experiment Setup

Our code for experiments on the GLUE benchmark is adopted from Malladi et al. (2023), including their dataset processing and prompt tuning workflow. Following prior work (Zhang et al., 2023; Malladi et al., 2023), we sample the perturbations from $\mathcal{N}(0, I_d)$ due to the concentration of measure in high-dimension and the empirical observations that sampling from $\mathcal{N}$ and $\mathcal{S}$ yields very similar performance (Malladi et al., 2023; Zhang et al., 2023).

Consistent with prior first-order SAM papers (Li et al., 2024), we use $\mathcal{U}(r\mathbb{B}^d)$ as the perturbation set with $r \in \{0.003, 0.005, 0.01, 0.03, 0.05\}$. We tune $t$ from $\{1,5,20\}$ and select the best one based on validation performance. We use $k=5$ for all SAM and TSAM experiments except for using $k=3$ for TSAM on the SQuAD-OPT experiment due to memory constraints.

We set $\rho=0.002$ for both RoBERTa and OPT and use $k=5$ for all zeroth-order methods. For TSAM and ZEST, we try $t=\{1,5,20\}$ and compare with $t=0$, which corresponds to ESAM (first-order) and MeZO (zeroth-order).

**RoBERTa-Base experiments.** All the first-order methods run for a maximum of 200 epochs, and all the zeroth-order ones run for a maximum of 700 epochs, with early stopping enabled. For SGD, SAM, ESAM, and TSAM, we tune the batch-size from $\{8,32\}$ and $\eta\in\{2e-3,1e-3,5e-4,2e-4,1e-4\}$. For MeZO and ZEST, we fix the batch-size as 128 and tune $\eta\in\{2e-5,1e-5,5e-6\}$.

**OPT-1.3B and LLaMA-7B experiments.** We run the first-order methods for maximally 30 epochs (or 3750 steps), and we run the zeroth-order ones for maximally 20K steps. Following the baseline (Malladi et al., 2023), we fix the batch-size to be 8 for first-order methods and 16 for zeroth-order ones. For SGD, we tune $\eta\in\{5e-5,1e-5,5e-6\}$; for SAM, ESAM, and TSAM, we tune $\eta\in\{5e-2,1e-2,1e-3\}$; for MeZO and ZEST, we tune $\eta\in\{5e-6,2e-6,1e-6\}$.

**ViT experiments.** All the first-order methods run for a maximum of 70 epochs, and all the zeroth-order ones run for a maximum of 400 epochs, with early stopping enabled. We fix the batch-size of all the methods to 64 and tune $\eta\in\{5e-3,2e-3,1e-3\}$ for first-order methods and $\eta\in\{1e-6,2e-6,5e-6\}$ for zeroth-order methods. We try $t\in\{1,5\}$ for simplicity.

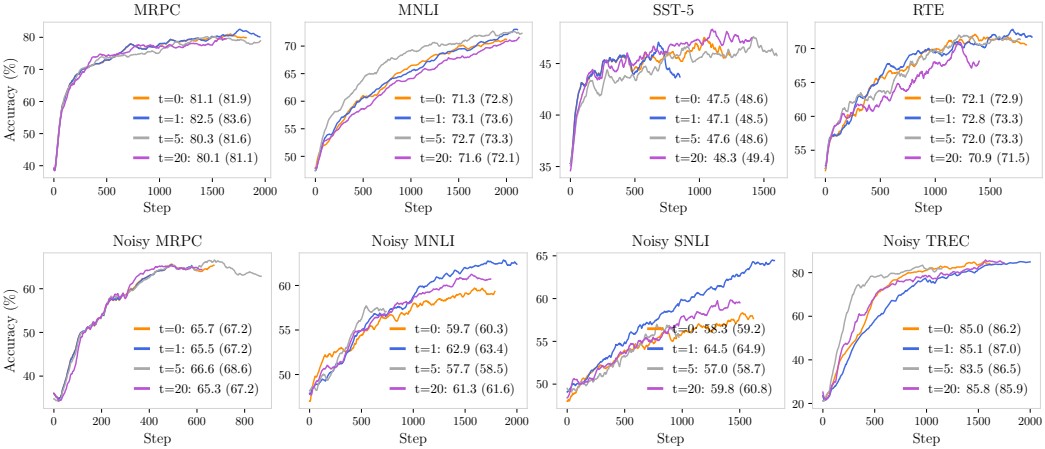

Figure 4: Validation accuracies of MeZO ($t=0$) and ZEST ($t=\{1,5,20\}$) on different datasets with clean labels (Upper) and 30% noisy labels (Bottom). The x-axis denotes evaluation steps. On each dataset, we have $k=5$ sampled perturbations. The curves are smoothed for visualization, so we report the final smoothed accuracy and the final raw accuracy written in the brackets. The results show that (1) there are multiple $t$ values that yield superior performance than MeZO and (2) $t=1$ almost always leads to superior performance to MeZO ($t=0$). It is noticeable that $t=1$ outperforms $t=0$ by 1.7%, 0.8%, 0.4%, 2.9%, 5.7%, and 0.8% in raw accuracies in the above plots; the only occasion when $t=1$ underperforms is on SST-5 by 0.1%.

## C.2 EXPERIMENT RESULTS

In this section, we present additional experiment results, including additional synthetic examples, the sharpness of the solutions found by different methods on more datasets, comparison with additional baselines, and an ablation study on $k$ (the number of sampled perturbations each iteration).

**Loss landscape of $t$-SAM.** We show that the loss landscape of $t$-SAM approaches that of SAM as $t$ increases in Figure 5, based on an example from Li et al. (2024). The leftmost figure is vanilla $f(x)$, whose global minimum is sharp. The second to the fourth figures correspond to $t$-SAM, $t=\{0,5,20\}$, respectively. It shows that (1) $t$-SAM losses approach SAM loss in the rightmost figure and (2) the average-loss SAM has the sharp minimum as solutions, while $t$-SAM and max-loss SAM have the flat minimum as solutions.

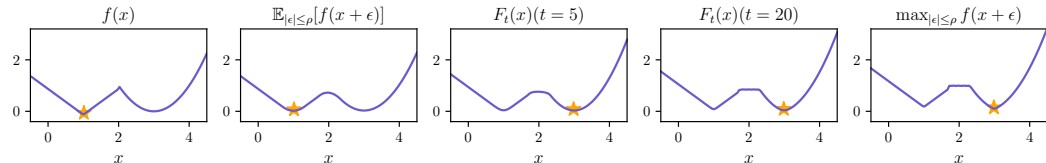

Figure 5: Loss landscapes of $f(x)$, average-loss SAM, $t$-SAM with $t=\{5,20\}$, and max-loss SAM. The original loss is $f(x)=|x-1|-0.13$ if $x\leq2$, and $f(x)=(x-3)^2$ otherwise. Therefore, $x=3$ is a flatter solution than $x=1$, though $f(1)<f(3)$. The orange star indicates the global minimum.

**More baselines.** To improve zeroth-order methods, prior work has leveraged momentum (Zhang et al., 2024) and Stochastic Variance-Reduced Gradient (SVRG) (Gautam et al., 2024) techniques in zeroth-order optimization. We compare with MeZO-SVRG, in which we evaluate the full batch gradient once an epoch, which corresponds to once every $q=\{4,8,20\}$ steps. From Table 7, we observe that ZEST outperforms MeZO-SVRG on 6/8 datasets.

We note that MeZO-SVRG requires storing one gradient vector in memory and requires twice the computation cost as MeZO, while ZEST preserves MeZO's memory and computational efficiency. Additionally, the contribution of ZEST is orthogonal to the techniques that can be applied to the classic zeroth-order methods, because we can also combine momentum and variance-reduction with our tilted gradients to further improve our performance.

Table 7: Experiments on RoBERTa-Base (512 training examples per class).

| Type | Task
Task type | SST-2
sentiment cls. | SST-5 | QQP
paraphrase | MRPC | TREC
topic cls. | MNLI
natural language inference | SNLI | RTE |
|---|---|---|---|---|---|---|---|---|---|
| 0th-
order | MeZO | 92.1 | 48.6 | 71.4 | 81.9 | 94.8 | 71.8 | 78.2 | 72.9 |
| | MeZO-SVRG | 91.5 | **50.3** | **74.5** | 78.7 | 87.2 | 70.3 | 70.7 | 66.8 |
| | ZEST$_N$ | **92.2** | 49.4 | 71.6 | **83.6** | **95.6** | 73.6 | **78.3** | **73.3** |
| | ZEST$_{BC}$ | 92.0 | 49.7 | 72.6 | 81.6 | 95.2 | **73.8** | 78.2 | 72.9 |

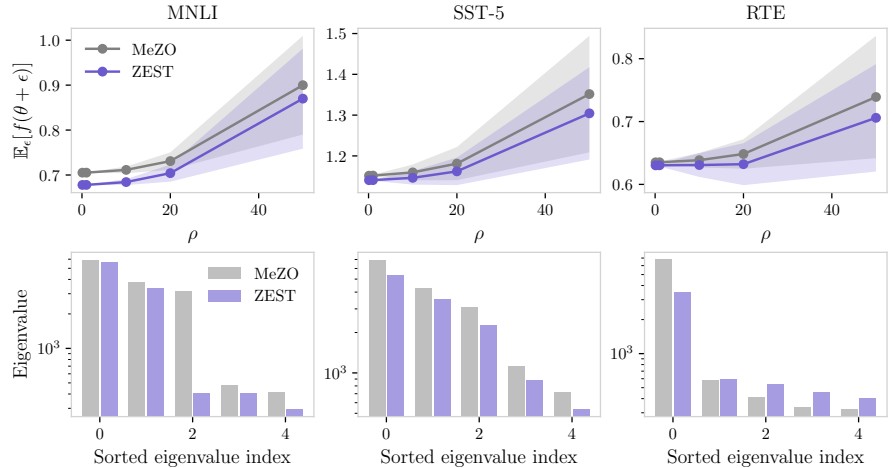

Figure 6: Sharpness of the solutions found by MeZO and ZEST on MNLI, SST-5, and RTE with RoBERTa-Base. Upper: Sharpness measured by $\mathbb{E}_{\|\epsilon\|\leq\rho}[f(x+\epsilon)]$. The scatters denote the average loss among 500 sampled perturbations and the shade denotes the standard deviation. Lower: Sharpness measured by the top-5 eigenvalues of $\nabla^2 f(x)$. The results suggest that ZEST yields flatter solutions in terms of both the robustness to parameter perturbations and largest curvature of the loss landscape at the arrived minimum, which agrees with our theoretical analysis in Section 4.

**Ablation study on $k$.** We perform an ablation study on $k$, the number of sampled perturbations each iteration. We report the performance of MeZO and ZEST on MNLI and MRPC with $k\in[5]$ in

Figure 7. We observe that the accuracy of both MeZO and ZEST on these two datasets increases monotonically as $k$ increases. Additionally, ZEST outperforms MeZO under any $k$.

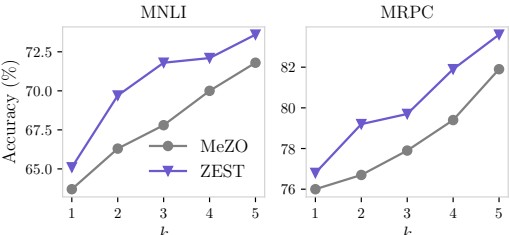

Figure 7: Validation accuracies of MeZO ($t$=0) and ZEST$_N$ ($t$={1,5}) w.r.t. the number of sampled perturbations each iteration ($k \in [5]$).

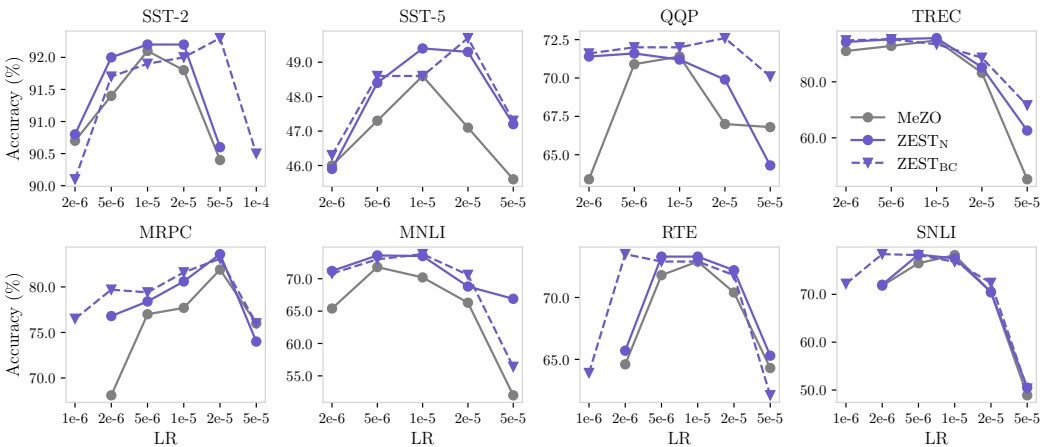

Figure 8: Validation accuracies of MeZO, ZEST$_N$, and ZEST$_{BC}$ with learning rates tuned separately. Training samples have clean labels.

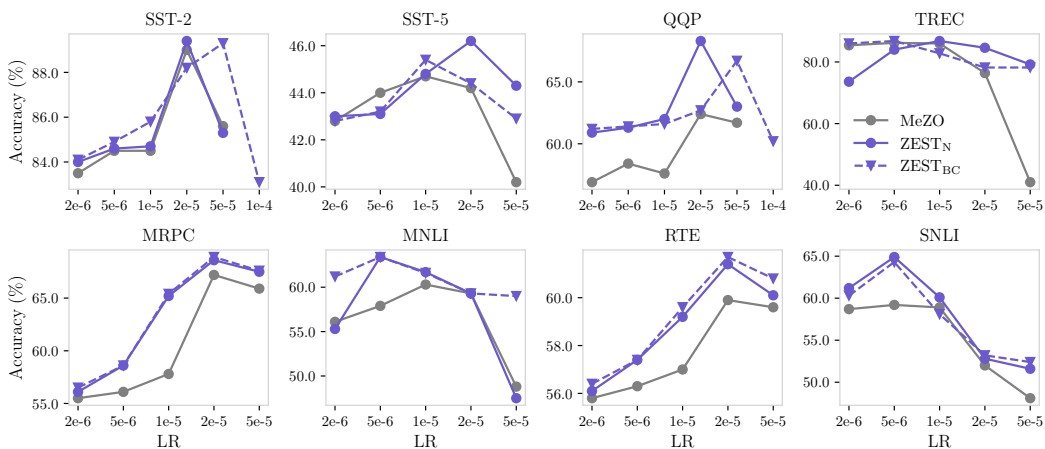

Figure 9: Validation accuracies of MeZO, ZEST$_N$, and ZEST$_{BC}$ with learning rates tuned separately. 30% of the training samples have noisy labels.

