# OpenReview forum: "Zeroth-Order Sharpness-Aware Learning with Exponential Tilting"
_ICLR.cc/2026/Conference — Submitted to ICLR 2026_

### Official Review · Reviewer_6vgJ · 2025-10-26

**Soundness:** 2
**Presentation:** 2
**Contribution:** 2
**Rating:** 4
**Confidence:** 4

**Summary:**

The paper proposes to apply t-SAM objective on top of MeZo to replace other SAM variants. Empirical studies show that it's effective and generalizes better.

**Strengths:**

1. This is a simple and straightforward idea and it's well motivated.
2. I enjoy reading about the insights from section 4.3.
3. Interesting ablation study on t

**Weaknesses:**

1. The empirical results seem very inconsistent across benchmarks against MeZo and doesn't support the claim of the paper. (table 2, table 3 and table 4)
2. limited novelty, it appears to me as a direct application of t-SAM objective on top of MeZo.

**Questions:**

1. Why does TSAM consistently outperform SAM and SGD, whereas the counterpart ZEST has mixed results?
2. Why is it necessary to compare noisy label? The gradient of MeZo itself is already very noisy. Would it be more informative if you can provide an ablation study on k (number of perturbations), comparing the two methods?

---

> ### Author Response · Authors · 2025-11-21
> **Rebuttal by Authors**
>
> We appreciate the reviewer for the comments, and we address the concerns and clarification questions as follows.
>
> **[Experiment results]** We clarify that ZEST outperforms MeZO consistently: Our method corresponds to purple-shaded rows in Table 2-4, which have higher accuracies/F1 scores than MeZO on every dataset. Our results show that sharpness-aware first-order methods outperform non-sharpness-aware first-order ones, and our sharpness-aware zeroth-order methods outperform weaker sharpness-aware zeroth-order approaches (MeZO). Even if we pick the lowest performance between our two variants ZEST$\_{\text{N}}$ and ZEST$\_{\text{BC}}$ (i.e., being unfair to ourselves), ZEST outperforms MeZO on 17/20 datasets, with an accumulative gap of 1.2\% summed over the remaining 3 datasets.
>
> **[Novelty]** We would like to clarify that we do not regard developing, analyzing, and evaluating zeroth-order algorithms for a sharpness-aware framework as straightforward. First, we lay out technical challenges and proofs of our algorithm in Appendix B in our original submission. It requires differential geometry to derive the true objective of running gradient descent with zeroth-order gradients under any $\rho$. In our proof, we leverage the divergence theorem and Stein’s lemma to derive the zeroth-order gradients (Eq. (3) and (4), proofs in Appendix B). Hence, we believe it is fundamentally different from simply applying two-point zeroth-order gradient estimation on top of the $t$-SAM objective, which would give biased estimates.
>
> Additionally, we propose using zeroth-order to solve more effective SAM variants, observing that classic zeroth-order methods inherently minimize a weak sharpness-aware objective. Our framework instantiates the classific zeroth-order method (and its sharpness notions) as a special case, and we theoretically quantify and illustrate why our proposed alternative sharpness notions (larger $t$ values) are better than the classific one ($t=0$). We have made this clearer by presenting our theoretical claims in formal theorems in the revision. From Theorem 4.1 and 4.2, we show that
> - When $t\to 0$, our sharpness recovers that of the average-loss SAM (and classic zeroth-order) objective, which can only detect “overall flatness” but not “flatness in all the directions”;
> - As $t$ increases, the penalty from each Hessian eigenvalue changes from uniformity to dominance by $\lambda_{\text{max}}$, so it allows us to choose how pessimistic we are when being sharpness-aware by changing $t$ values;
> - As $t\to\infty$, ours recovers the sharpness notion of the max-loss SAM objective under ball perturbation (where this loss is well-defined).
>
> Further, we use a systematic set of experiments to support our claim: classic zeroth-order methods inherently minimize a weak sharpness-aware objective, and our ZEST, which solves more effective SAM objectives, can generalize better in a wider range of practical problems.
>
> **[Noisy label experiments]** Being robust to label noise is one of the most prominent gains for sharpness-aware approaches [1,4], and it is common to evaluate on datasets with noisy labels in SAM literature [2,3]. The reviewer is correct that the gradient of zeroth-order methods is noisy, which biases them to find flat minima [5]. Our contribution presents that the sharpness notion of MeZO does not particularly penalize the sharpest direction as ZEST does. Performance on datasets with noisy labels demonstrates this by showing that ZEST outperforms MeZO by 0.4% to 5.9% in accuracy.
>
> **[Ablation study on $k$]** We thank the reviewer for suggesting an ablation study on $k$ (number of perturbations each iteration). We report the performance of MeZO and ZEST on MNLI and MRPC with $k\in[5]$ below:
>
> |Method|k|MRPC|MNLI|
> |:--|:--:|:--:|--:|
> |MeZO|1|76.0|63.7|
> ||2|76.7|66.3|
> ||3|77.9|67.8|
> ||4|79.4|70.0|
> ||5|81.9|71.8|
> |ZEST$\_{\text{N}}$|1|76.6|65.1|
> ||2|79.2|69.7|
> ||3|79.7|71.8|
> ||4|81.9|72.1|
> ||5|83.6|73.6|
>
> We observe that the accuracy of both MeZO and ZEST on these two datasets increases monotonically as $k$ increases. Additionally, ZEST outperforms MeZO under any $k$. We have added and visualized these ablation results in Figure 7 of our revised paper.
>
> [1] Why is sam robust to label noise? ICLR 2024.
> [2] Sharpness-aware minimization for efficiently improving generalization. ICLR 2021.
> [3] Tilted sharpness-aware minimization. ICML 2025.
> [4] An Analysis of Concept Bottleneck Models: Measuring, Understanding, and Mitigating the Impact of Noisy Annotations. NeurIPS 2025.
> [5] Zeroth-Order Optimization Finds Flat Minima. NeurIPS 2025.

---

> > ### Comment · Reviewer_6vgJ · 2025-11-26
> > **Further question on the inconsistency of empirical results**
> >
> > For example, in table 2, the lower end of ZEST
> >
> > loses on SST-2, MRPC
> >
> > ties on SNLI, RTE
> >
> > wins on SST-5, QQP,  TREC, MNLI
> >
> > so I would appreciate a bit of insights on why the two ZEST variants can't consistently outperform MeZO?
> > Under which circumstances and which one should be preferred?

---

> > ### Comment · Reviewer_6vgJ · 2025-11-26
> > **Additional question on empirical experiments**
> >
> > what is the methodology of adjusting learning rates, perturbation strength as well as other hyperparameters? Maybe I missed it somewhere in the paper, but ZEST has very different update norm compared to MeZO due to the normalization.

---

> > > ### Author Response · Authors · 2025-11-30
> > > **Response**
> > >
> > > We appreciate the reviewer’s engagement with our rebuttal and the questions. Our response to each of the further questions is as follows:
> > >
> > > **[Hyperparameter tuning]** Please refer to our Appendix C.1 of the original submission or revision for the learning rate (LR), batch size (BS), and perturbation scale ($\rho$) that we used/tried. Specifically, the proper range of each hyperparameter is determined based on MeZO's, which is then applied to ZEST.
> > >
> > > - All zeroth-order methods use the same BS that we determine based on the MeZO codebase: Our OPT experiments use the same number of training samples as MeZO, so we use BS=16 as they do; In RoBERTa experiments, MeZO code uses BS=64 for 16-shot, so we use BS=128 for our 512-shot.
> > > - All zeroth-order methods use the same $\rho$, and the proper $\rho$ for each (model, dataset) pair is picked as the one that yields the best convergence of MeZO on SST-2. Then we use this $\rho$ for all the ZEST experiments and on other datasets.
> > > - With the above BS and $\rho$ chosen, we first try a coarse LR search on MeZO on {1e-3, 1e-4, 1e-5, 1e-6} for RoBERTa. Finding that 1e-5 is the best, we try the fine-grained LR search on {5e-5, 2e−5, 1e−5, 5e−6}. Finding 5e-5 bad for MeZO, we use the range {2e−5, 1e−5, 5e−6} for ZEST experiments and find that it works well.
> > >
> > > Overall, our hyperparameter tuning was first conducted on MeZO and then applied to ZEST, so it is overly generous to the baseline methods (i.e., a bit unfair to ourselves). Theoretically, we clarify that “normalization” mentioned by the reviewer is also needed in MeZO gradient, which is in the form of dividing by $n$ in Algorithm 2 of MeZO [1] (line $\theta_i \leftarrow \theta_i - (\eta_t/n) \text{projected\\_grads}[j] * z$; their $n$ is our $k$, i.e., number of perturbations). We note that the gradient of MeZO and ZEST differ just by whether it is uniform or a weighted average of gradients of perturbed losses:
> > >
> > > The objective of MeZO is $F_0(x)=\mathbb{E}_u[f(x+\rho u)]$, whose gradient is
> > > $$ \nabla_x F_0(x)=\nabla_x \mathbb{E}_u[f(x+\rho u)] = \mathbb{E}_u[\nabla_xf(x+\rho u)]. $$
> > > The objective of ZEST is $F_t(x)=\frac{1}{t} \log \left( \mathbb{E}_u[ e^{tf(x+\rho u)}] \right)$, whose gradient is
> > > $$ \nabla_x F_t(x)= \frac{\mathbb{E}_u[e^{tf(x+\rho u)}\nabla_x f(x+\rho u)]}{\mathbb{E}_u[e^{tf(x+\rho u)}]}. $$
> > >
> > > When we sample $k$ perturbations each iteration, we have
> > > $$ \nabla_{\text{MeZO}} = \sum_{i=1}^k \frac{1}{k}  \nabla_x f(x+\rho u_i)$$
> > > and
> > > $$ \nabla_{\text{ZEST}} = \frac{\frac{1}{k}\sum_{i=1}^k e^{tf(x+\rho u_i)}\nabla_x f(x+\rho u_i)}{\frac{1}{k}\sum_{i=1}^k e^{tf(x+\rho u_i)}} = \sum_{i=1}^k w_{t,i} \nabla_x f(x+\rho u_i) \quad \text{where} \quad w_{t,i}=\frac{e^{tf(x+\rho u_i)}}{\sum_{j=1}^k e^{tf(x+\rho u_j)}}.$$
> > > This shows that the MeZO gradient is the average of the gradients of perturbed losses, and the ZEST gradient is a weighted average by $w_{t,i} \in (0,1)$ with $\sum_{i} w_{t,i} =1$.
> > >
> > > **[Table 2]** If we compare MeZO’s accuracy with min\{ZEST$\_{\text{N}}$, ZEST$\_{\text{BC}}$\}, we have a -0.1\% on SST-2, -0.3\% on MRPC, but ties or wins on other datasets. The similar trend also holds for Table 3: min\{ZEST$\_{\text{N}}$, ZEST$\_{\text{BC}}$\} outperforms MeZO on 7 out of 8 tasks. Since these previous results were obtained by applying MeZO’s proper learning rate range to ZEST, we further conducted fair learning rate tuning for each method separately and present the following results. In the following table, we highlight a ZEST accuracy if it outperforms MeZO. We observe that under fair learning rate tuning, both ZEST$\_\text{N}$ and ZEST$\_\text{BC}$ consistently outperform MeZO. The new search space is {1e-4, 5e-5, 2e-5, 1e-5, 5e-6, 2e-6, 1e-6}, which ensures that the best LR does not occur on the boundary of the range. We present the accuracy of each method under each learning rate in Figure 8-9 of our revision.
> > >
> > > Table: Best accuracies of each method with their learning rate tuned separately.
> > > |Data label|Method|SST-2|SST-5|QQP|MRPC|TREC|MNLI|SNLI|RTE|
> > > |:--|:--:|:--:|:--:|:--:|:--:|:--:|:--:|:--:|--:|
> > > |clean|MeZO|92.1|48.6|71.4|81.9|94.8|71.8|78.2|72.9|
> > > ||ZEST$\_{\text{N}}$|**92.2**|**49.4**|**71.6**|**83.6**|**95.6**|**73.6**|**78.3**|**73.3**|
> > > ||ZEST$\_{\text{BC}}$|**92.3**|**49.7**|**72.6**|**83.1**|**95.2**|**73.8**|**78.4**|**73.5**|
> > > |30\% noisy|MeZO|89.0|44.7|62.4|67.2|86.2|60.3|59.2|59.9|
> > > ||ZEST$\_{\text{N}}$|**89.4**|**46.2**|**68.3**|**68.6**|**86.8**|**63.4**|**64.9**|**61.4**|
> > > ||ZEST$\_{\text{BC}}$|**89.3**|**45.4**|**66.7**|**68.9**|**86.8**|**63.4**|**64.3**|**61.7**|
> > >
> > > [1] Fine-Tuning Language Models with Just Forward Passes. NeurIPS 2023.
> > > [2] Sharpness-aware minimization for efficiently improving generalization. ICLR 2021.

---

> > > ### Author Response · Authors · 2025-11-30
> > > **Response (cont.)**
> > >
> > > We thank the reviewer's engagement with our rebuttal again and continue presenting our responses as follows.
> > >
> > > **[ZEST$\_\text{N}$ vs. ZEST$\_\text{NC}$]** In Figure 3 in our revision, we show that the bias-corrected estimator indeed results in smaller bias. We observe that (1) both estimators correctly converge to the ground truth (i.e., error norm reduces to 0) as $k$ increases, (2) the bias-corrected variant has smaller bias when $k$ is finite, which agrees with the theory [3].
> > >
> > > However, it is worth noting that these two estimators have approximately the same variance (the shades in Figure 3, and it is proved in [3]). In practice, when we sample a small number of random seeds due to efficiency considerations, the performance of two estimators can be similar. This is why ZEST$\_{\text{BC}}$ performs similarly to our ZEST$\_{\text{N}}$, though the former has a smaller bias in theory.
> > >
> > > Despite this, we introduce both estimators to illustrate that our method is general and can admit various ratio estimators. Additionally, when there are ample computational resources to run a large number of trials such as in low-dimensional problems, it is favorable to leverage bias correction. We have added related discussions in Section 3.2, Section 5.2, and Appendix B.4.
> > >
> > > [3] Mean and variance of ratio estimators used in fluorescence ratio imaging. Cytometry: The Journal of the International Society for Analytical Cytology.

---

### Official Review · Reviewer_myiW · 2025-10-26

**Soundness:** 2
**Presentation:** 3
**Contribution:** 2
**Rating:** 4
**Confidence:** 2

**Summary:**

This paper considers the Tilted Sharpness-Aware Minimization (t-SAM) objective and applies the zeroth-order optimization technique to solve this optimization problem. The t-SAM objective has a special structure and the format of the zeroth-order gradient estimator presents its challenge in obtaining an unbiased estimation. To address this issue, this paper further proposes the bias-correction method. The empirical results validate the performance of these approaches and validate the improved generalization ability of minimizing t-SAM objective using zeroth-order method.

**Strengths:**

This paper has made solid LLM experimental results, which include sufficient baselines in a famous LLM training baseline over multiple commonsense reasoning datasets. The results are expected; minimizing SAM will improve the generalization ability and the performance in test set indeed validates this statement. Also, this paper has included a new bias correction approach for the gradient estimation in minimizing t-SAM. This result is new as in the standard optimizaiton problem, the zeroth-order gradient estimation is unbiased when the step is sufficiently small.

**Weaknesses:**

1. The main contribution of this work seems to be evaluating the zeroth-order gradient for the t-SAM objective function. As the t-SAM objective is given from another paper, the theoretical contribution of this work seems to be not strong.

2. The bias correction seems to sound. However, I would prefer to see some synthetic examples showing that the bias is indeed corrected.

3. Given that the theoretical contribution is not strong, I may expect to see more large sacle experiments. However, the current experiments are taken on small LLMs.

**Questions:**

Given these weaknesses, I may have following questions:

1. In introduction, the author claims "We show that our method can identify and conservatively avoid minima with large curvatures in any direction, while vanilla methods cannot (Section 4)." Is Section 4 a formal theorem or just some intuitive explainations on why it can identify and conservatively avoid minima with large curvatures?

2. I hope the author could further clarify the main theoretical contributions. Evaluating the zeroth-order gradient for the t-SAM objective function given in Theorem 3.1 seems to be too weak as its only theoretical contribution.

3. I would encourage the author to include additional visualization on the bias of each gradient in Theorem 3.1; also, it would be better if we may see the bias is scaled in the rate $O(1/\sqrt{d})$ as commented at the bottom on page 3.

4. I am actually concerned about the correctness of the $O(1/\sqrt{d})$ rate. Has it been proved somewhere?

5. I would also encourage the author to include some vision tasks just as the t-SAM objective's paper *Tilted Sharpness-Aware Minimization* did.

6. I am not very sure how the parameter $t$ is chosen. When $t$ tends to infinite, the objective function becomes the desired SAM objective function. Does it mean that we shold make $t$ as large as possible?  However, in Appendix B.8, we cannot make $t$ too large. Do I misunderstand something?

---

> ### Author Response · Authors · 2025-11-21
> **Rebuttal by Authors**
>
> We are grateful for the reviewer’s detailed feedback. We present our responses and address the concerns as follows.
>
> **[Theoretical contributions]** Our theoretical contributions are twofold. First, we develop zeroth-order algorithms for a general sharpness-aware objective, where we lay out technical challenges and proofs in Appendix B. We leverage the divergence theorem and Stein’s lemma to derive the **unbiased** zeroth-order gradients for the objective under arbitrary $\rho$ (Eq. (3) and (4), proofs in Appendix B). Hence, we believe it is fundamentally different from simply applying two-point zeroth-order gradient estimation on top of the $t$-SAM objective, which would give biased estimates.
>
> Second, we theoretically characterize the spectrum of the sharpness notion of $t$-SAM, which includes the sharpness notions of the classic methods as a special case. We define Regularizer Sensitivity (Definition 4.1) to quantify the increased penalty from the sharpest directions as $t$ increases, and we visualize low-dimensional examples to demonstrate how different values of $t$ impact the favored solutions during optimization. We have made this clearer by presenting our theoretical claims in formal theorems in the revision. From Theorem 4.1 and 4.2, we show that
> - When $t\to 0$, our sharpness recovers that of the average-loss SAM (and classic zeroth-order) objective, which can only detect “overall curvature flatness” but not “flatness in all the directions”;
> - As $t$ increases, the penalty from each Hessian eigenvalue changes from uniformity to dominance by $\lambda_{\text{max}}$, so it allows us to choose how pessimistic we are when being sharpness-aware by changing $t$ values;
> - As $t\to\infty$, ours recovers the sharpness notion of max-loss SAM objective under ball perturbation (where this loss is well-defined).
>
> Further, we use a systematic set of experiments to support our claim: classic zeroth-order methods inherently minimize a weak sharpness-aware objective, and our ZEST, which solves more effective SAM objectives, can generalize better in a wider range of practical problems.
>
> **[Experiments on large LLM]** We further present experimental results of LLaMA-7B on SuperGLUE datasets and extend our OPT-1.3B experiments to more SuperGLUE datasets, including WIC and WSC. The results agree with our main texts, suggesting that (1) TSAM and SAM are superior to ESAM, and (2) ZEST outperforms MeZO and matches/outperforms first-order methods on multiple datasets.
>
> Table 1: Performance of OPT-1.3B on additional SuperGLUE datasets.
> |Type|Method|WSC|WIC|
> |:--|:--:|:--:|--:|
> |1st|SGD|57.8|65.2|
> ||ESAM|58.8|66.8|
> ||SAM|64.6|**68.7**|
> ||TSAM|**66.5**|67.1|
> |0th|MeZO|61.5|55.5|
> ||ZEST$\_{\text{N}}$|**64.4**|57.1|
> ||ZEST$\_{\text{BC}}$|63.5|**57.2**|
>
> Table 2: Performance of LLaMA-7B on four SuperGLUE datasets.
> |Type|Method|COPA|ReCROD|WSC|WIC|
> |:--|:--:|:--:|:--:|:--:|--:|
> |1st|SGD|85.0|82.4|61.5|66.5|
> ||ESAM|85.0|82.5|62.2|65.7|
> ||SAM|**86.0**|**82.8**|**63.5**|**67.9**|
> ||TSAM|OOM|OOM|OOM|OOM|
> |0th|MeZO|**89.0**|80.1|60.8|64.4|
> ||ZEST$\_{\text{N}}$|**89.0**|**81.8**|62.7|**66.2**|
> ||ZEST$\_{\text{BC}}$|**89.0**|81.3|**63.4**|65.9|
>
> **[Formal statements in Section 4]** We have proved the claim the reviewer cited in the original paper in Section 4.1 and 4.2 of our original submission, hence it is a formal statement, as opposed to intuitive explanations. In our revision, we re-organized related paragraphs and presented those formal claims as Theorems 4.1 and 4.2.
>
> Specifically, we decompose the t-SAM objective into $f(x)+R_t(x)$, where $R_t$ is the sharpness regularizer. In Theorems 4.1 and 4.2 of our revision, we show that as $t$ increases, under both Gaussian and uniform ball perturbation, large Hessian eigenvalues contribute more to $R_t$ than small Hessian eigenvalues. Therefore, ZEST with $t\neq0$ biases against next-steps with large curvatures in any direction. In contrast, when $t=0$, each eigenvalue contributes the same to $R_t$, so the classic zeroth-order method can only detect “overall flatness” but not “flatness in all the directions.” Empirically, we visualize a low-dimensional example in Section 4.3 in our original submission. It shows that an increased $t$ leads to bias toward local minima with flat curvature in any direction, which agrees with our theory.

---

> > ### Author Response · Authors · 2025-11-21
> > **Rebuttal by Authors (cont.)**
> >
> > We thank the reviewer for detailed feedback again and continue presenting our responses as follows.
> >
> > **[Bias-corrected estimator]** We are glad that the reviewer finds the bias correction variant technically sound. In Figure 3 in our revision, we show that the bias-corrected estimator indeed results in smaller bias.  We report the bias and variance of two estimators on a toy problem across different $k$ values. We observe that (1) both estimators correctly converge to the ground truth (i.e., error norm reduces to 0) as $k$ increases, (2) the bias-corrected variant has smaller bias when $k$ is small, which agrees with theory [2].
> >
> > **[Bias of using $\mathbb{E}_{\mathcal{S}}$ in denominator]** Quantifying the bias boils down to finding how sampling from $\text{Unif}(\mathcal{B})$ is different from $\text{Unif}(\mathcal{S})$. Because uniformly sampling $v$ from the ball in contrast to the sphere differs only in the expected norm of $v$, given that the directions are both uniformly distributed. Since the expected norm $\mathbb{E}[\Vert v\Vert] = \sqrt{d}$ and its standard deviation is $\text{std}(\Vert v\Vert)=O(1/\sqrt{d})$, we know that when $d$ is large, the difference in norm is negligible. Please refer to Appendix B.3 in our original submission for its proof.
> >
> > We also explicitly quantify the bias between Eq. (4) and (5) as follows, with proof in Appendix B.3 in our revision:
> >
> > Assume that $f(x;\xi)$ is $L$-smooth and $M$-Lipschitz for any $x,\xi$. We have two estimators
> > $$ \nabla(x) = \frac{1}{t\rho} \frac{\underbrace{\mathbb{E}\_{\mathcal{S}}[(e^{tf(x+\rho u)} - e^{tf(x-\rho u)})u]}\_{N\_{\mathcal{S}}}}{\underbrace{\mathbb{E}\_{\mathcal{B}}[e^{tf(x+\rho v)} + e^{tf(x-\rho v)}]}\_{D\_{\mathcal{B}}}} \quad \text{and} \quad \tilde{\nabla}(x) = \frac{1}{t\rho} \frac{N\_{\mathcal{S}}}{\underbrace{\mathbb{E}\_{\mathcal{S}}[e^{tf(x+\rho u)} + e^{tf(x-\rho u)}]}\_{D\_{\mathcal{S}}}},$$
> >
> > and the bias from replacing $D_{\mathcal{B}}$ with $D_{\mathcal{S}}$ in the denominator is
> > $$
> >     \text{Bias}(x) = \nabla(x) - \tilde{\nabla}(x) = \frac{N_{\mathcal{S}}}{t\rho} \left(\frac{1}{D_{\mathcal{B}}} - \frac{1}{D_{\mathcal{S}}}\right) = \frac{N_{\mathcal{S}}}{t\rho} \left(\frac{D_{\mathcal{S}} - D_{\mathcal{B}}}{D_{\mathcal{B}} D_{\mathcal{S}}}\right).
> > $$
> > With choosing $\rho \leq d^{-\frac{4}{5}}$, we have
> > $$\Vert \text{Bias}(x) \Vert \leq O\left(\frac{1}{\sqrt{d}}\right).$$
> >
> > **[Experiments on vision tasks]** We thank the reviewer for this suggestion. We finetune a pretrained small ViT model on CIFAR-10 with both clean and 30% noisy labels, following prior work [3]. The following table suggests that (1) more pessimistic sharpness notions are beneficial since SAM and TSAM consistently outperform SGD and ESAM, and (2) ZEST outperforms MeZO.
> > |Type|Method|Clean|Noisy|
> > |:--|:--:|:--:|--:|
> > |1st|SGD|96.9|95.1|
> > ||ESAM|97.5|95.7|
> > ||SAM|**97.9**|**97.5**|
> > ||TSAM|**97.9**|**97.4**|
> > |0th|MeZO|80.2|69.0|
> > ||ZEST$\_{\text{N}}$|**82.8**|**72.3**|
> > ||ZEST$\_{\text{BC}}$|82.4|71.8|
> >
> > **[Performance under big $t$]** We note that SAM (corresponding to $t\to\infty$) is a min-max objective that can be too pessimistic and difficult to solve because it is non-smooth. It considers the worst-case loss in the neighborhood, whereas there can be many bad points in the neighborhood that incur large losses that need to be considered. Theoretically, it is an open question what the best sharpness characterization is that leads to the best generalization (not necessarily the max eigenvalue of Hessian as in SAM), since the existing generalization bounds are data-dependent and there are no executable principles [1,3,4]. Similar to previous work [1], we empirically show the effects of $t$ and we suggest that $t=1$ is a safe go-to choice since it almost always yields superior performance to MeZO in different settings. We have clarified this in Appendix B.8 of our revision.
> >
> > [1] Tilted sharpness-aware minimization. ICML 2025.
> > [2] Mean and variance of ratio estimators used in fluorescence ratio imaging. Cytometry: The Journal of the International Society for Analytical Cytology.
> > [3] Generalization and robustness of the tilted empirical risk. ICML 2025.
> > [4] How does sharpness-aware minimization minimize sharpness? ICLR 2023.

---

> > > ### Comment · Reviewer_myiW · 2025-11-23
> > >
> > > I appreciate the comprehensive rebuttal. I am satisfied with the added experiments and the justification on the $1/\sqrt{d}$ bias.
> > >
> > > **[Theoretical contributions for Theorem 3.1]** The author replied "We leverage the divergence theorem and Stein’s lemma to derive the **unbiased** zeroth-order gradients for the objective under arbitrary $\rho$ (Eq. (3) and (4), proofs in Appendix B). " . However,
> > > 1. I don't consider Eq. (3) and (4) as **unbiased** zeroth-order gradients. As the author has done in Sec. 3.2, these formulas are only asymptotically unbiased, and their bias would be $1/k^2$ if using bias correction.
> > > 2. Moreover, the actual estimation constructed in Eq.(6) is simply estimating the numerator and dominator of the gradient of t-SAM objective, respectively. The gradient formula is known by using the original t-SAM's paper.
> > > 3. The bias correction is also standard for estimating the expectation in the form A/B.
> > >
> > > So, I am still not convinced that Theorem 3.1 would be anything novel contribution.
> > >
> > > **[Theoretical contributions for Section 4]** I understand by using different $t$, the results in Section 4 will recover different sharpness. Both Theorem 4.1 and Theorem 4.2 are related to this behavior of $R_t$ under different $t$. However,
> > > 1. These two results are not surprising as the t-SAM objective function already has this property (when t=0, it reduces to the naive one, when t=infity, it reduces to another).
> > > 2. For this paper, the connection between the Sharpness Sensitivity and the Flatness of the minima has not been fully justified. I am still unclear how it claims "conservatively avoid minima with large curvatures in any direction, while vanilla methods cannot".

---

> > > > ### Author Response · Authors · 2025-11-25
> > > > **Response**
> > > >
> > > > We thank the reviewer for additional comments, and we are glad to know that the reviewer is satisfied with the added experiments and clarification on the $O(1/\sqrt{d})$ rate. Our response to each of the further questions is as follows.
> > > >
> > > > **[Theoretical contributions of Theorem 3.1]**
> > > >
> > > > 1. In our previous response, by "unbiased" we meant that we derive an exact gradient formula using only zeroth-order access to the model (Eq. (3) and (4)). We agree that our proposed approximation (Eq. (5)-(7)) in our actual algorithms are only asymptotically unbiased, as we noted in the paper and as the reviewer mentioned. We thank the reviewer for this feedback, and we have further clarified this (asymptotically unbiasedness of our estimator) in the paper. Additionally, we add an ablation study on $k$ (number of perturbations each iteration) to showcase the empirical impact of $k<\infty$. The performance of MeZO and ZEST on MNLI and MRPC with $k\in[5]$ is reported below:
> > > >
> > > >    |Method|k|MRPC|MNLI|
> > > >    | :-- | :--: | :--: | :--: |
> > > >    | MeZO|1|76.0|63.7|
> > > >    ||2|76.7|66.3|
> > > >    ||3|77.9|67.8|
> > > >    ||4|79.4|70.0|
> > > >    ||5|81.9|71.8|
> > > >    |ZEST$_{\text{N}}$|1|76.6|65.1|
> > > >    ||2|79.2|69.7|
> > > >    ||3|79.7|71.8|
> > > >    ||4|81.9|72.1|
> > > >    ||5|83.6|73.6|
> > > >
> > > >    We observe that the accuracy of both MeZO and ZEST on these two datasets increases monotonically as $k$ increases. Additionally, ZEST outperforms MeZO under any $k$. We have added and visualized these ablation results in Figure 7 of our revised paper.
> > > >
> > > > We agree that the first-order gradient of $t$-SAM appears in prior literature, which we didn't claim as our contribution. Our contributions include deriving the zeroth-order gradient of $t$-SAM, which has not been proposed in prior work. It is not as straightforward as other potential baselines that may not result in the exact gradient estimates as Eq. (3) and (4), e.g., simply plugging in $t$-SAM objective to the classic two-point estimator. We also agree that plug-in and bias-corrected estimators are existing methods in prior literature. Part of our contribution in algorithm design is to allow the flexibility of incorporating any ratio estimators in the ZEST algorithm. We show these two options are easy to implement and achieve competitive performance on diverse tasks.
> > > >
> > > > Given the reviewer's comments, we are happy to revise the statement of Theorem 1 to convey the message better.
> > > >
> > > > **[Theoretical contributions of Section 4]**
> > > >
> > > > 1. We note that while the $t$-SAM paper [1] studies properties of their objectives, it did not explicitly derive the sharpness notions as we do in our paper. In addition, we'd like to clarify that our theorems explore the sharpness notions for a family of objectives under both ball and Gaussian perturbations, where we just highlight the two extreme cases as special cases. Thus, our explicit characterization of sharpness is novel.
> > > > 2. We decompose the $t$-SAM loss as $f(x)+R_t(x)$, where $f(x)$ is the classic average risk and $R_t$ is the regularizer (i.e., sharpness).
> > > >    - When $t\to 0$, the regularizer is $R_{\text{avg}} \propto \sum_i^d\lambda_i$ (paper line 1127 and 1167). This means that each eigenvalue contributes *equally* to the regularizer (and thus the loss) because $\partial R_{\text{avg}}/\partial \lambda_i= \text{constant}$. Therefore, MeZO optimizes for small "loss + average curvature", not "loss + the worst curvature".
> > > >    - As $t$ increases, we show that $\partial R_t/\partial \lambda_i$ is a function of $\lambda_i$ that increases as $\lambda_i$ increases (e.g., paper line 289). This means that large Hessian eigenvalues are more weighted in their contribution to $R_t$ than small Hessian eigenvalues. As eigenvalues of Hessian represent curvature along the corresponding eigenvector, our result (e.g., line 289) implies that we penalize large curvatures in different, non-uniform ways compared with prior work. Prior MeZO update penalizes the average of the eigenvalues; SAM (approximately) focuses on the largest eigenvalue; We prove that our algorithm is penalizing all eigenvalues, where each eigenvalue $\lambda_i$ is weighted differently based on the value of $\lambda_i$ itself. If $t$ keeps increasing, we have $R_t$ approximately proportional to (and dominated by) a few largest eigenvalues (e.g., the equation in line 289 shows that the sensitivity increases as t increases).
> > > >    - The above can be illustrated by the example in Figure 1 (b), where the loss landscape has two local minima that have the same loss values and $\sum_i^d \lambda_i$ but different $\lambda_{\max}$. MeZO (i.e., ZEST with $t=0$) treats two local minima the same, but ZEST with $t>0$ converges to the minimum whose $\lambda_{\max}$ is smaller (i.e., curvature is small in all directions).
> > > >
> > > > [1] Tilted sharpness-aware minimization. ICML 2025.

---

> > > > > ### Comment · Reviewer_myiW · 2025-11-26
> > > > >
> > > > > Appreciate it for the detailed response. I remain my concerns on the theoretical contributions, so I hope maintain my current score. However, I am not fighting to reject this paper. If other reviewers are positive and advocate for acceptance, I am happy to be overruled.

---

> > > > > > ### Author Response · Authors · 2025-11-26
> > > > > > **Response**
> > > > > >
> > > > > > We sincerely appreciate your engagement with our response. Could you please elaborate on your concern about our theoretical contributions given our response above? We are happy to provide any further clarification that might be helpful.

---

### Official Review · Reviewer_YGy9 · 2025-10-30

**Soundness:** 3
**Presentation:** 3
**Contribution:** 3
**Rating:** 6
**Confidence:** 4

**Summary:**

The paper proposes ZEST, a zeroth-order sharpness-aware optimization method that uses exponential tilting to interpolate between average-loss and worst-case SAM objectives. The approach introduces a curvature-sensitive regularizer that encourages flatter minima while remaining fully gradient-free and memory-efficient. Experiments show consistent improvements over MeZO with the comparable computational cost.

**Strengths:**

1. The paper presents a unified framework that bridges zeroth-order optimization and sharpness-aware minimization (SAM) through exponential tilting. This idea provides a continuous spectrum between average-loss and worst-case objectives.

2. Experiments show consistent improvements over MeZO.

**Weaknesses:**

1. The experiments are conducted only on relatively small models. These are modest compared to current state-of-the-art LLMs (e.g., LLaMA-7B, Mistral-7B, OPT-6.7B). As a result, it would be more convincing if the proposed ZEST framework scales to modern large-parameter settings.

2. The evaluation mainly focuses on classic datasets such as GLUE, SQuAD, and ReCoRD. These benchmarks are saturated and may not reflect the difficulty or diversity of current NLP tasks. It would strengthen the paper to include contemporary datasets such as SuperGLUE, MMLU.

**Questions:**

How does ZEST perform compared with other zeroth order methods such as those in [1] and [2]?
[1] Variance‑reduced Zeroth‑Order Methods for Fine‑Tuning Language Models (Gautam et al., 2024)
[2] Revisiting Zeroth‑Order Optimization for Memory‑Efficient LLM Fine‑Tuning: A Benchmark (Zhang et al., 2024)

**Details Of Ethics Concerns:**

None.

---

> ### Author Response · Authors · 2025-11-21
> **Rebuttal by Authors**
>
> We are grateful for the reviewer’s careful reading and valuable comments. We present our responses as follows.
>
> **[Experiments on large models and more datasets]** We agree with the reviewer that it would strengthen the paper to include more datasets and models. We extend our OPT-1.3B experiments to more SuperGLUE datasets, including WIC and WSC. We also add experiments for LLaMA-7B on SuperGLUE datasets. The results in the following tables agree with our main texts, suggesting that (1) TSAM and SAM are superior to ESAM, and (2) ZEST outperforms MeZO and matches/outperforms first-order methods on multiple datasets.
>
> Table 1: Performance of OPT-1.3B on additional SuperGLUE datasets.
> |Type|Method|WSC|WIC|
> |:--|:--:|:--:|--:|
> |1st|SGD|57.8|65.2|
> ||ESAM|58.8|66.8|
> ||SAM|64.6|**68.7**|
> ||TSAM|**66.5**|67.1|
> |0th|MeZO|61.5|55.5|
> ||ZEST$\_{\text{N}}$|**64.4**|57.1|
> ||ZEST$\_{\text{BC}}$|63.5|**57.2**|
>
> Table 2: Performance of LLaMA-7B on four SuperGLUE datasets.
> |Type|Method|COPA|ReCROD|WSC|WIC|
> |:--|:--:|:--:|:--:|:--:|--:|
> |1st|SGD|85.0|82.4|61.5|66.5|
> ||ESAM|85.0|82.5|62.2|65.7|
> ||SAM|**86.0**|**82.8**|**63.5**|**67.9**|
> ||TSAM|OOM|OOM|OOM|OOM|
> |0th|MeZO|**89.0**|80.1|60.8|64.4|
> ||ZEST$\_{\text{N}}$|**89.0**|**81.8**|62.7|**66.2**|
> ||ZEST$\_{\text{BC}}$|**89.0**|81.3|**63.4**|65.9|
>
> **[Compare with more baselines]** We thank the reviewer for pointing out these related work. We have added the references and discussions in Section 2 and Appendix C.2 in the revised paper. We note that the variance reduction technique, adding momentum, etc., are orthogonal to our algorithm, where we propose new methods for estimating the zeroth-order gradient of a different objective. We can also combine momentum or variance-reduction with our tilted gradients to further improve the performance of our method. Having said that, we directly compare our current algorithm with MeZO-SVRG [1]: We evaluate the full batch gradient once an epoch (i.e., every 8/12/20 steps) and find that ZEST outperforms MeZO-SVRG on 6/8 datasets with RoBERTa-Base. We note that MeZO-SVRG requires storing one gradient vector in memory and requires twice the computation cost as MeZO, while ZEST preserves MeZO’s memory and computational efficiency.
>
> |Method|SST-2|SST-5|QQP|MRPC|TREC|MNLI|SNLI|RTE|
> |:--|:--:|:--:|:--:|:--:|:--:|:--:|:--:|--:|
> |MeZO|92.1|48.6|71.4|81.9|94.8|71.8|78.2|72.9|
> |MeZO-SVRG|91.5|**50.3**|**74.5**|78.7|87.2|70.3|70.7|66.8|
> |ZEST$\_{\text{N}}$|**92.2**|49.4|71.6|**83.6**|**95.6**|73.6|**78.3**|**73.3**|
> |ZEST$\_{\text{BC}}$|92.0|49.7|72.6|81.6|95.2|**73.8**|78.2|72.9|
>
> [1] Variance-reduced Zeroth-Order Methods for Fine-Tuning Language Models. ICML 2024.

---

### Official Review · Reviewer_2S7G · 2025-11-11

**Soundness:** 3
**Presentation:** 3
**Contribution:** 2
**Rating:** 4
**Confidence:** 5

**Summary:**

This paper proposes a new zeroth-order optimization algorithm (ZEST) that uses exponential tilting technique to recover the smooth spectrum of sharpness-aware objectives. Particularly, it considers the tilted sharpness-aware minimization (t-SAM) objective, which is the smooth approximation of the SAM objective. When $t\to \infty$, this objective function tends to the standard SAM objective. Then this work uses the divergence theorem to approximate t-SAM gradients and obtain the tilted zeroth-order gradient estimator. Then naive plug-in and bias-corrected plug-in are proposed to obtain this derived estimator, leading to the ZEST algorithm. Further sharpness analysis is presented to provide the explicit bias of the t-SAM objective under two common perturbations. Lastly, experiments are conducted on RoBERTa models to validate the theoretical findings and the empirical performances of the ZEST algorithm.

**Strengths:**

1. This paper provides a simple and clear formula for the tilted zeroth-order gradient. It utilizes the structure of the gradient of t-SAM objective.

2. The sharpness analysis provides a solid foundation for the soundness of the proposed approach. The explicit dependence of $R_t$ on the relevant parameters deepens the understanding on this new algorithm.

3. The authors validate ZEST on a diverse set of tasks (classification, QA, generation) and models (RoBERTa, OPT). The consistent improvements over the MeZO baseline , especially on noisy data, demonstrate the method's effectiveness.

**Weaknesses:**

1. The core t-SAM objective is not novel to this work; it is adopted from Li et al. (2024) . Adapting this objective to the zeroth-order setting is good but not the same as inventing the objective.

2. The theoretically superior "Bias-Corrected" estimator does not consistently outperform the "Naive" estimator in Table 2 and Table 3. It weakens the motivation of designing the biased-corrected plug-in.

**Questions:**

1. As the bias-corrected plug-in does not consistently outperform the naive plug-in in these experiments, does the bias-correction simply not provide a significant benefit? What is the point of proposing this method?

2. In Algorithm 1, the update step in Line 11 is written inside the loop over $i=1,2,\dots,k$. This implies the model parameter $x$ is updated $k$ times per iteration. Should the update in Algorithm 1 be a single step using the sum (i.e., $x\leftarrow x - \eta \frac{t\rho} \sum_i w_i v_i$)?

3. When $t=20$, the t-SAM objective is more closed to the SAM objective but its performance shown in Figure 3 is not better (in Noisy MNLI and Noisy SNLI). Is there any intuitive explanation on this  phenomenon?

4. In Figure 3, why are these curves not in the same length.

5. Can the authors quantify the theoretical gap between the t-SAM objective and the original SAM objective? A simple example validating how the t-SAM solution approaches the SAM solution as $t$ increases would be helpful.

---

> ### Author Response · Authors · 2025-11-21
> **Rebuttal by Authors**
>
> We appreciate the reviewer’s time and detailed feedback. We address the concerns and clarification questions as follows.
>
> **[Contribution]** We thank the reviewer for acknowledging our contribution to elucidating the sharpness-aware nature of zeroth-order optimization, under different sharpness notions. We clarify that our contribution is not to propose a new sharpness-aware objective, but to (1) identify that the classic zeroth-order method is inherently sharpness-aware but its sharpness notion is not optimal in many cases, (2) leverage $t$-SAM and derive its zeroth-order gradient under arbitrary $\rho$, (3) characterize the sharpness notions as a function of $t$ and illustrate why $t>0$ is superior to $t=0$ (i.e., classic zeroth-order method), and (4) evaluate our method on a wide range of model and task types.
>
> **[Motivation of bias-corrected estimator]**  In Figure 3 in our revision, we show that the bias-corrected estimator indeed results in smaller bias.  We report the bias and variance of two estimators on a toy problem across different $k$ values. We observe that (1) both estimators correctly converge to the ground truth (i.e., error norm reduces to 0) as $k$ increases, (2) the bias-corrected variant has smaller bias when $k$ is small, which agrees with theory [3].
>
> However, note that these two estimators have approximately the same variance [3]. In practice, when we sample a small number of random seeds due to efficiency considerations, the performance of two estimators can be similar. This is why ZEST$\_\text{BC}$ performs similarly to our ZEST$\_\text{N}$, though the former has a smaller bias in theory.
>
> Despite this, we introduce both estimators to illustrate that our method is general and can admit various ratio estimators. Additionally, when there are abundant computational resources to run many trials, it is favorable to leverage bias correction. We have added related discussions in Section 3.2, Section 5.2, and Appendix B.4.
>
> **[Update step clarification]** We'd like to clarify that our update rule is essentially (following the reviewer's notations)  $x \leftarrow x-\frac{\eta}{t\rho}w_1v_1-\cdots-\frac{\eta}{t\rho}w_kv_k$, which is equivalent to $x \leftarrow x-\frac{\eta}{t\rho} \sum_i w_iv_i$, as suggested by the reviewer. We present it in that particular way (Line 10-12) to highlight the procedure of memory-efficient implementation: For each $i$, we delete $v_i$ from memory after it is used to perturb model parameters, and we need to re-generate them by the same random seed one-by-one when we make updates. This trick is commonly used in prior work [1] (Algorithm 2 of MeZO).  To implement the single-step update, one needs to store $v_i, i\in[k]$ in memory, which can be expensive. We have clarified this in Section 3.2 of our revision.
>
> **[Performance under big $t$]** We note that SAM is a min-max objective that can be too pessimistic and difficult to solve because it is non-smooth. It considers the worst-case loss in the neighborhood, whereas there can be many bad points in the neighborhood that incur large losses that need to be considered. Theoretically, it is an open question what the best sharpness characterization is that leads to the best generalization (not necessarily the max eigenvalue of Hessian as in SAM), since the existing generalization bounds are data-dependent and there are no executable principles [2,4]. Similar to prior work [2], we empirically show the effects of $t$ and we suggest $t=1$ as a safe go-to choice since it almost always yields superior performance to MeZO across settings. We have clarified this in Appendix B.8 of our revision.
>
> **[Curves in Figure 3 (now Figure 4 in revision)]** We clarify that we employ early-stopping for all the zeroth-order methods if there is no improvement in validation loss for 200 iterations. The plots show smoothed accuracy values for cleaner visualization, which might smooth out the peak accuracy that triggers the early-stopping later.
>
> **[$t$-SAM solution example]** We thank the reviewer for this suggestion. In Figure 5 of our revision, we visualize how the loss landscape of $t$-SAM approaches that of SAM as $t$ increases, based on an example from prior work. The leftmost figure is vanilla $f(x)$, whose global minimum is sharp. The second to the fourth figures correspond to $t$-SAM loss with $t=\\{0,5,20\\}$, respectively. It shows that (1) $t$-SAM loss landscapes approach the vanilla SAM loss as $t$ increases and (2) the original loss and average-loss SAM have the sharp minimum as solutions, while $t$-SAM and max-loss SAM have the flat minimum as solutions.
>
> [1] Fine-Tuning Language Models with Just Forward Passes. NeurIPS 2023.
> [2] Tilted sharpness-aware minimization. ICML 2025.
> [3] Mean and variance of ratio estimators used in fluorescence ratio imaging. Cytometry: The Journal of the International Society for Analytical Cytology.
> [4] Generalization and robustness of the tilted empirical risk. ICML 2025.

---

### Meta-Review · Area_Chair_PsgP · 2026-01-07

**Summary:**

All reviewers except one voted to reject this paper. On one hand they appreciated the unified framework and experiments of the paper. On the other they had issues with (a) small models, (b) old datasets, and (c) contribution of the paper. The authors report results for larger models and newer datasets, resolving (a) and (b). However, they do not convince the reviewers that the theoretical contributions are significant enough to the community to justify acceptance. Given that the experimental results when comparing 1st and 0th order methods are mixed, the paper would benefit from a clearer experimental and use-case justification of 0th order methods. If this could be done the paper would be a much stronger sell. For these reasons, I vote to reject the current version of the paper.

**Reviewer Concerns:**

Please see above.

**Reviewer Scores:**

I believe reviewers would have reduced their score or kept it the same.

---

### Decision · Program_Chairs · 2026-01-26

Reject